# CRISPR-based herd immunity can limit phage epidemics in bacterial populations

Pavel Payne[1,2]*, Lukas Geyrhofer[3], Nicholas H Barton[2], Jonathan P Bollback[1,2]*

[1]Institute of Integrative Biology, University of Liverpool, Liverpool, United Kingdom; [2]Institute of Science and Technology Austria, Klosterneuburg, Austria; [3]Department of Chemical Engineering, Technion - Israel Institute of Technology, Haifa, Israel

**Abstract** Herd immunity, a process in which resistant individuals limit the spread of a pathogen among susceptible hosts has been extensively studied in eukaryotes. Even though bacteria have evolved multiple immune systems against their phage pathogens, herd immunity in bacteria remains unexplored. Here we experimentally demonstrate that herd immunity arises during phage epidemics in structured and unstructured *Escherichia coli* populations consisting of differing frequencies of susceptible and resistant cells harboring CRISPR immunity. In addition, we develop a mathematical model that quantifies how herd immunity is affected by spatial population structure, bacterial growth rate, and phage replication rate. Using our model we infer a general epidemiological rule describing the relative speed of an epidemic in partially resistant spatially structured populations. Our experimental and theoretical findings indicate that herd immunity may be important in bacterial communities, allowing for stable coexistence of bacteria and their phages and the maintenance of polymorphism in bacterial immunity.

DOI: https://doi.org/10.7554/eLife.32035.001

*For correspondence:
pavel.payne@liverpool.ac.uk (PP);
J.P.Bollback@liverpool.ac.uk (JPB)

**Competing interests:** The authors declare that no competing interests exist.

## Introduction

The term 'herd immunity' has been used in a variety of ways by different authors (see *Fine et al., 2011*). Here, we define it as a phenomenon where a fraction of resistant individuals in a population reduces the probability of transmission of a pathogen among the susceptible individuals. Furthermore, if the fraction of resistant individuals in a population is sufficiently large the spread of a pathogen is suppressed. Experimental research into the phenomenon has focused mostly on mammals (*Jeltsch et al., 1997*; *Mariner et al., 2012*), birds (*van Boven et al., 2008*; *Meister et al., 2008*), and invertebrates (*Konrad et al., 2012*; *Wang et al., 2013*). In human populations the principles of herd immunity were employed to limit epidemics of pathogens through vaccination programs (*Fine et al., 2011*), which in the case of smallpox lead to its eradication between 1959 and 1977 (*Fenner, 1993*).

Alongside advances in vaccination programs, the formalization of a general theory of herd immunity was developed. The theory is based on a central parameter, $R_0$, which describes the fitness of the pathogen, as measured by the number of subsequent cases that arise from one infected individual in a population (for a historical review of $R_0$ see [*Heesterbeek, 2002*]). Thus, $R_0$ indicates the epidemic spreading potential in a population. Given $R_0$ the herd immunity threshold is defined as,

$$H = \frac{R_0 - 1}{R_0}, \qquad (1)$$

which determines the required minimum fraction of resistant individuals needed to halt the spread of an epidemic. $R_0$ and subsequently also $H$ are affected by the specific details of transmission and the contact rate among individuals (*Grassly and Fraser, 2008*). Many theoretical studies have addressed the influence of some of these details, in particular maternal immunity (*Anderson and*

**eLife digest** When a disease spreads through a population, it encounters certain individuals it cannot infect. If there are enough of these individuals, the epidemic stops. This phenomenon is known as 'herd immunity', and it occurs in many animals – for example, it plays an important role in human vaccination schemes.

While bacteria can cause disease, they are themselves targeted by viruses called 'phages'. Bacteria can overcome this threat, and they resist phage attacks in ways that are well understood at the molecular level. However, little is known about the impact of this resistance at the scale of the population. Can herd immunity occur in bacteria? If so, what factors influence the threshold at which it will occur? In other words, what affects the minimum percentage of immune bacteria needed to stop the spread of a phage infection?

To answer these questions, Payne et al. used both experimental and mathematical methods. For the experiments, a phage and two strains of bacteria were used, one immune to the virus and one not. The two strains were combined to form several populations with different percentages of resistant bacteria, and the phage was added. How the virus could spread in these different populations was recorded. This confirmed that herd immunity does occur in bacteria and showed how the resistant bacteria influence the speed which an epidemic spreads in a population.

Building on the experiments, Payne et al. then produced a mathematical model to explore how different factors affect herd immunity. For example, the model showed that the thresholds for herd immunity can be predicted from how quickly bacteria and phages replicate. The thresholds are lower when bacteria reproduce more quickly, but higher when it is the phages that multiply faster.

The model also helps infer a formula that informs on how diseases spread in any species, such as humans. In particular, it becomes possible to predict herd immunity thresholds based on how quickly an epidemic spreads in a population where few people are vaccinated. Future research is needed to adapt the formula to the specific factors that shape disease outbreaks in humans. Ultimately, this could help policymakers design strategies to deal with infectious diseases, such as yearly outbreaks of the flu.

DOI: https://doi.org/10.7554/eLife.32035.002

*May, 1992-08*), age at vaccination (*Anderson and May, 1982*; *Nokes and Anderson, 1988*), age related or seasonal differences in contact rates (*Schenzle, 1984*; *Anderson and May, 1985*; *Yorke et al., 1979*), social structure (*Fox et al., 1971*), geographic heterogeneity (*Anderson and May, 1984*; *Lloyd and May, 1996*; *Real and Biek, 2007*), and the underlying contact network of individuals (*Ferrari et al., 2006*).

Interestingly, little work has focused on the potential role of herd immunity in microbial systems which contain a number of immune defense systems and have an abundance of phage pathogens. These defenses vary in their potential to provide herd immunity as they target various stages of the phage life cycle, from adsorption to replication and lysis. Early defense mechanisms include the prevention of phage adsorption by blocking of phage receptors (*Nordström and Forsgren, 1974*), production of an extracellular matrix (*Hammad, 1998*; *Sutherland et al., 2004*), or the excretion of competitive inhibitors (*Destoumieux-Garzón et al., 2005*). Alongside these bacteria have evolved innate immune systems that target phage genomes for destruction. These include host restriction-modification systems (RMS) (*Blumenthal and Cheng, 2002*), argonaute-based RNAi-like systems (*Swarts et al., 2014*), and bacteriophage-exclusion (BREX) systems (*Goldfarb et al., 2015*). In addition to innate systems, bacteria have evolved an adaptive immune system called CRISPR-Cas (clustered regularly interspaced short palindromic repeat) (*Sorek et al., 2013*). In order for any of these immune systems to provide herd immunity, they must prevent further spread of the pathogen. Therefore, unless the phage particles degrade in the environment at a timescale comparable to the phage adsorption rate, the immune system must provide a 'sink' for the infectious particles reducing the average number of successful additional infections below one. Unlike the early defense mechanisms that may simply prevent an infection but not the further reproduction of infectious particles, the RMS, BREX, argonaute-based RNAi-like, and the CRISPR-Cas systems degrade foreign phage DNA after it is injected into the cell, and thus continue to remove phage particles from the

environment, which increases their potential to provide herd immunity. In order for herd immunity to arise, the population must also be polymorphic for immunity, which can be achieved if immunity is plasmid borne. In addition to this, the CRISPR-Cas system is unique in that it is adaptive allowing cells to acquire immunity upon infection (see *Figure 1A,B and C*), which can lead to polymorphism in immunity even if the system is chromosomal.

In addition to immune system-specific factors, the reproductive rate of phage depends strongly on the physiology of the host bacterium (*Hadas et al., 1997*), and the underlying effective contact network which may vary greatly in bacterial populations depending on the details of their habitat. Thus, herd immunity will be influenced by the physiological state of the bacteria and the mobility of the phage in the environment through passive diffusion and movement of infected individuals. Taken together these details call into question the applicability of the traditional models of herd immunity from vertebrates to phage-bacterial systems. Thus, experimental investigation and further development of extended models that take into account the specifics of microbial systems are required.

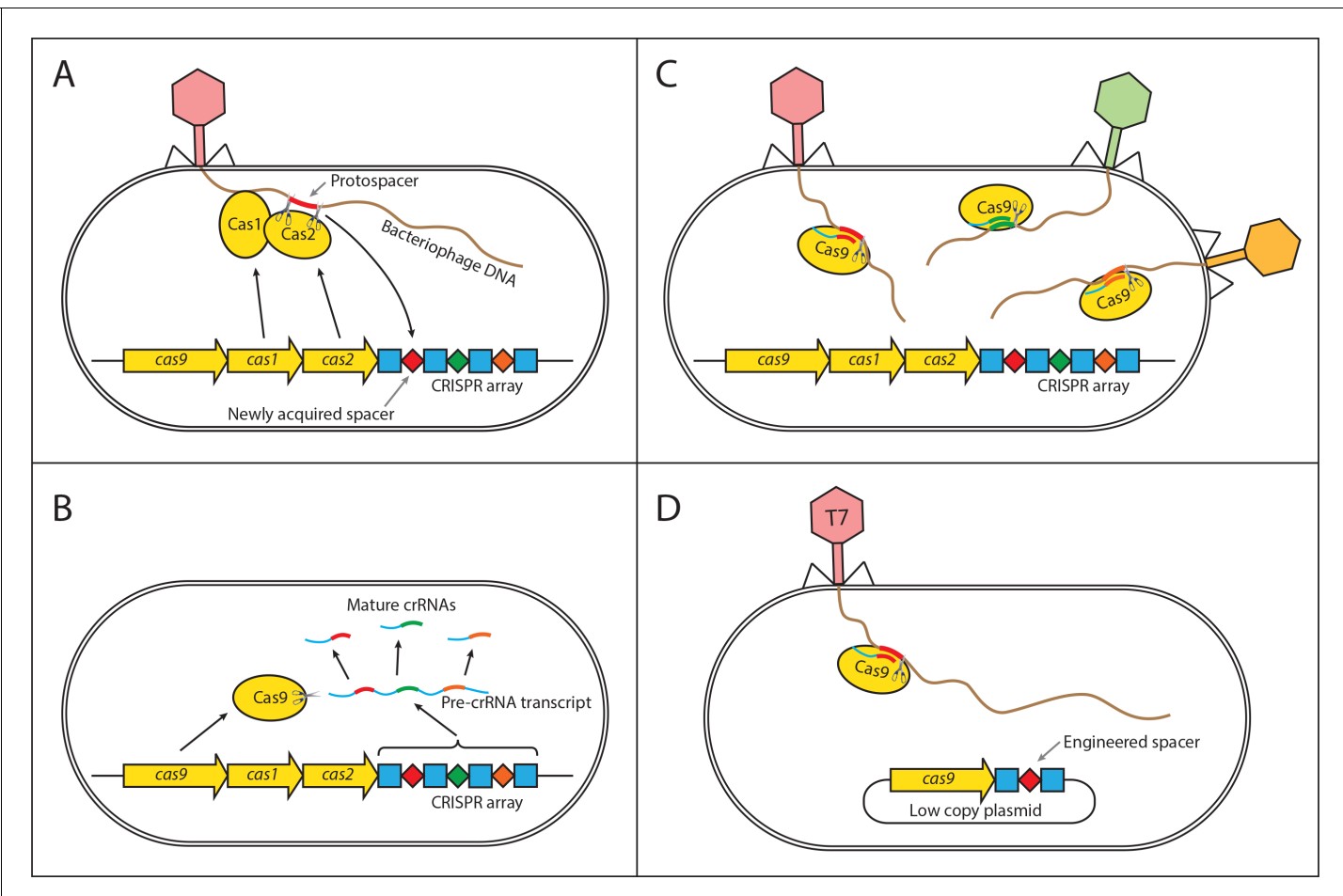

**Figure 1.** Mechanism of CRISPR/Cas type II immunity. The CRISPR/Cas system provides immunity to phages and its main features can be described by three distinct stages. (**A**) Acquisition. When a cell gets infected by a phage, a protospacer on the invading phage DNA (indicated as a red bar) is recognized by Cas1 and Cas2. The protospacer is cleaved out and ligated to the leader end (proximal to the Cas genes) of the CRISPR array as a newly acquired spacer (red diamond). (**B**) Processing. The CRISPR array is transcribed as a Pre-crRNA and processed by Cas9 (assisted by RNaseIII and trans–activating RNA, not shown) into mature crRNAs. (**C**) Interference. Mature crRNAs associate with Cas9 proteins to form interference complexes which are guided by sequence complementarity between the crRNAs and protospacers to cleave invading DNA of phages whose protospacers have been previously incorporated into the CRISPR array. (**D**) A truncated version of the CRISPR system on a low copy plasmid, which was used in this study lacks cas1 and cas2 genes and was engineered to target a protospacer on the T7 phage chromosome to provide Escherichia coli cells with immunity to the phage. The susceptible strain contains the same plasmid except the spacer does not target the T7 phage chromosome.
DOI: https://doi.org/10.7554/eLife.32035.003

To investigate under which conditions herd immunity may arise in bacterial populations, we constructed an experimental system consisting of T7 phage and bacterial strains susceptible and resistant to it. Our experimental system can be characterized by the following features. First, we used two strains of Escherichia coli, one with an engineered CRISPR-based immunity to the T7 phage, and the other lacking it (*Figure 1D*). Second, we examined the dynamics of the phage spread in different environments – spatially structured and without structure. Furthermore, we developed and analyzed a spatially explicit model of our experimental system to determine the biologically relevant parameters necessary for bacterial populations to exhibit herd immunity.

## Results

### Properties of resistant individuals

We engineered a resistant E. coli strain by introducing the CRISPR-Cas Type II system from Streptococcus pyogenes with a spacer targeting the T7 phage genome (see Material and Methods). We further characterized the ability of the system to confer resistance to the phage. We find a significant level of resistance as measured by the probability of cell burst when exposed to T7 (*Figure 2A*). However, resistance is not fully penetrant as approximately 1 in 1000 resistant cells succumb to infection. In addition, we observe that as phage load increases (multiplicity of infection, MOI) the probability that a cell bursts increases (*Figure 2A*). In order to determine the herd immunity threshold in our experimental system, we constructed the resistant strain such that upon infection the cell

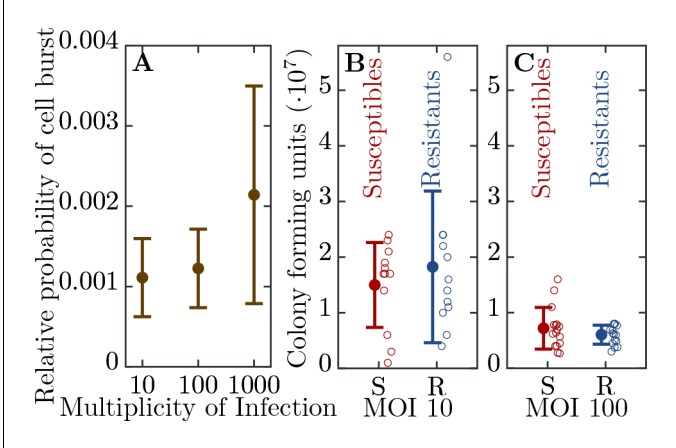

**Figure 2.** Efficiency of bacterial resistance. (**A**) The probability that a resistant cell bursts, relative to a susceptible cell, at three different initial multiplicities of infection (MOI). The probability that a resistant cell bursts at MOI 1000 is significantly higher than at MOI 10 ($p = 0.019$, $t_4 = 3.031$) or at MOI 100 ($p = 0.022$, $t_5 = 2.674$). The error bars show the standard deviations from the mean. Note that this measure is not a widely used 'efficiency of plating' but it determines the probability of burst of single resistant cells (see Materials and methods for details). (**B**) The number of colony forming units (CFUs) post phage challenge (see Materials and methods). The mean number of CFUs after the bacterial cultures were exposed to the phage is not significantly different between susceptible and resistant strains at MOI 10 ($p = 0.239$, $t_{22} = 0.721$) and (**C**) at MOI 100 ($p = 0.27$, $t_{30} = 1.124$), indicating that the resistant cells' growth is halted after the cells are infected by a phage. The error bars show the standard deviations from the mean. There were no detectable CFUs in either susceptible or resistant cell cultures at MOI 1000. It should be noted that the indicated MOI values do not correspond to the average number of phages that adsorb to cells in the experiments. For MOI 10 we estimated the mean number of phages per cell as 0.229 and for MOI 100 as 0.988 (see Materials and methods for details). It was impossible to determine the mean for MOI 1000 as there were no detectable CFUs under such conditions. The data presented in this figure can be found in *Figure 2—source data 1*.

DOI: https://doi.org/10.7554/eLife.32035.004

The following source data is available for figure 2:

**Source data 1.** Efficiency of bacterial resistance.

DOI: https://doi.org/10.7554/eLife.32035.005

growth is halted, yet the cell still adsorbs and degrades phages (*Figure 2B,C*). This feature is important as it prevents the action of frequency dependent selection which in naturally growing populations will favor the resistant strain until its frequency reaches the herd immunity threshold. Thus, in our system if the frequency of the resistant strain is below the herd immunity threshold, the resistant cells remain below the threshold and are unable to stop the epidemic and the whole population collapses. In contrast, if the frequency of resistant individuals in the population is above the herd immunity threshold, the resistant individuals provide complete herd immunity and the population survives. These properties allow us to quantify the expanding epidemic in both liquid media and on bacterial lawns (without and with spatial structure, respectively) using high throughput techniques. Specifically, it allows us to control for the complex dynamics of the system arising from frequency dependent selection and simultaneous changes in the physiological states of the cells (growth rates depending on the nutrient concentrations) and phage (burst size, latent period depending on the cell's physiology).

It should be noted that our model does not reflect this artificial property – it assumes that resistant bacteria keep growing after successfully overcoming a phage infection (see *Equation (2d)*). This discrepancy, however, does not affect the model prediction of the herd immunity threshold in our experimental system for the following reason: time scale of an epidemic spread through a population (double exponential phage growth) is substantially shorter than the time scale of bacterial population growth (exponential growth). Therefore, whether or not an epidemic is established does not depend on later dynamics of frequencies of resistant and susceptible individuals in the population, it only depends on the initial conditions. Similarly, the model correctly captures the dynamics of an epidemic in spatially structured populations as the phage spreads radially and in every time-point the epidemic front encounters a naive population with a constant ratio of resistant to susceptible individuals.

## Herd immunity in populations without spatial structure

To understand the influence of spatial population structure, or lack thereof, we first measured the probability of population survival (i.e., whether the cultures are cleared or not) in well mixed liquid environments (no spatial structure) consisting of differing proportions of resistant to susceptible individuals and T7 phage. When the percentage of resistant individuals is in excess of 99.6% all 16 replicate populations survive a phage epidemic (i.e., show no detectable difference in growth profiles to the phage free controls; *Figure 3*). Populations with 99.2% and 98.4% resistant individuals show intermediate probabilities of survival – 10 out of 16 replicate populations and 4 out of 16 replicate populations survive, respectively (*Figure 3*). The likely explanation as to why some populations survive and others collapse is due to the stochastic nature of phage adsorption after inoculation: If the population composition is close to the herd immunity threshold a stochastic excess of phage particles adsorbing to susceptible cells may trigger an epidemic, whereas if chance increases the number of phages adsorbing to resistant individuals, the epidemic is suppressed. However, when populations have fewer than 96.9% resistant individuals all 16 replicate populations fail to survive and collapse under the epidemic (*Figure 3*).

As mentioned in the introduction, phage and bacterial physiology may affect the herd immunity threshold. To test this we altered bacterial growth by reducing the concentration of nutrients in the medium by mixing LB broth with 1X M9 salts in different ratios (*Figure 4*), which concurrently alters the T7 phage's latent period and burst size (*Figure 5A,B* and *Table 1*). Indeed, we observe as bacterial growth rates decline the fraction of resistant individuals necessary for population survival decreases (*Figure 5C*). When the populations are grown in a 50% diluted growth medium, the fraction of resistant individuals required for a 100% probability of survival is 99.2%; when the fraction of resistant individuals is 75% or less populations go extinct. In a 20% growth medium the fraction of resistant individuals required for survival decreases to 96.9%, while the fraction when all replicates collapse to 50%.

From the experimental observations of the herd immunity threshold values we infer the phage $R_0$ using *Equation 1*. In an undiluted growth medium the phage $R_0$ falls between 32 and 256 and decreases to between 4 and 128 in 50% and between 2 and 32 in 20% nutrient medium. These data indicate that bacterial populations can exhibit herd immunity in homogeneous liquid environments. However, bacteria typically live in spatially structured environments such as surfaces, biofilms or

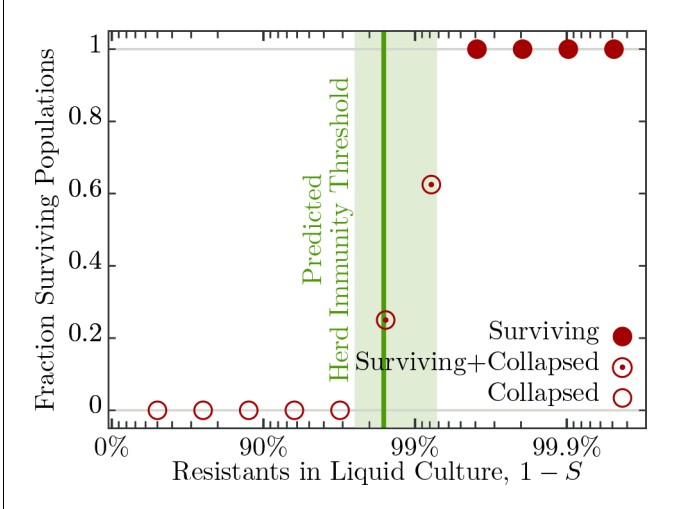

**Figure 3.** Fraction of surviving populations at $18\,h$ post phage infection. Bacterial populations consisting of various fractions of resistant to susceptible individuals infected with $\approx 50$ phages, corresponding to a multiplicity of infection (MOI) of $\approx 10^{-4}$, designed to resemble an epidemic initiated by the burst size from one infected individual (see *Table 2* for burst size estimates). Each population phage challenge is replicated 16 times. The solid dark green line shows the model prediction, *Equation 4*, for the herd immunity threshold ($H$), given latent period ($\lambda$), bacterial growth rate ($\alpha$), and phage burst size ($\beta$). Shaded area indicates $\pm 1$ standard deviation. The data presented in this figure can be found in *Figure 3—source data 1*.

DOI: https://doi.org/10.7554/eLife.32035.006

The following source data is available for figure 3:

**Source data 1.** Fraction of surviving populations at 18 hr post phage infection.

DOI: https://doi.org/10.7554/eLife.32035.007

micro-colonies, therefore we extended our experiments to consider the potential impact of spatially structured populations.

## Herd immunity in spatially structured populations

In order to discern the role, if any, spatial structure plays in herd immunity we conducted a set of experiments in spatially structured bacterial lawns on agar plates. Spatially structured bacterial populations provide a more fine grained measure of herd immunity, compared to the population survival assays done in liquid culture. On bacterial lawns, phages spread radially from a single infectious phage particle and the radius of plaque growth on different proportions of resistant to susceptible individuals can be easily quantified. In addition, these data allow for estimating the speed of the epidemic wave front in these different regimes using real-time imaging (*Figure 6A*).

We observe a decline in the number of plaque forming units (see *Appendix 2—figure 1*) and a significant decrease in final plaque sizes as the proportion of resistant individuals in the populations increases (*Figure 6B,C*). A reduction in the final plaque size compared to a fully susceptible population was statistically significant with as few as 10% resistant individuals in a population ($p = 0.004$, $t_{53} = 2.744$). In order to determine the effect of resistant individuals during the earlier phase of bacterial growth (until the bacterial density on the agar plate reaches saturation; *Figure 4A*), we analyze the velocities of plaque growth between 0 and 24 hr post inoculation (*hpi*). We find that the speed is significantly reduced after 11 *hpi* when the population consists of as few as 10% of resistant individuals ($p = 0.0317$, $t_{32} = 1.923$). As the fraction of resistant individuals further increases, the speed declines significantly at earlier and earlier time points: 6 *hpi* with 20% ($p = 0.0392$, $t_{62} = 1.79$), and 5.67 *hpi* with 30% ($p = 0.0286$, $t_{53} = 1.943$). In fact, when the fraction of resistant individuals exceeds 40%, the reduction in the speed of the spread is statistically significant immediately after the plaques are visually detectable (*Figure 7*). It should be noted that all populations with such low percentages of resistant individuals in liquid environment collapsed, indicating that spatial structure plays a role in herd immunity.

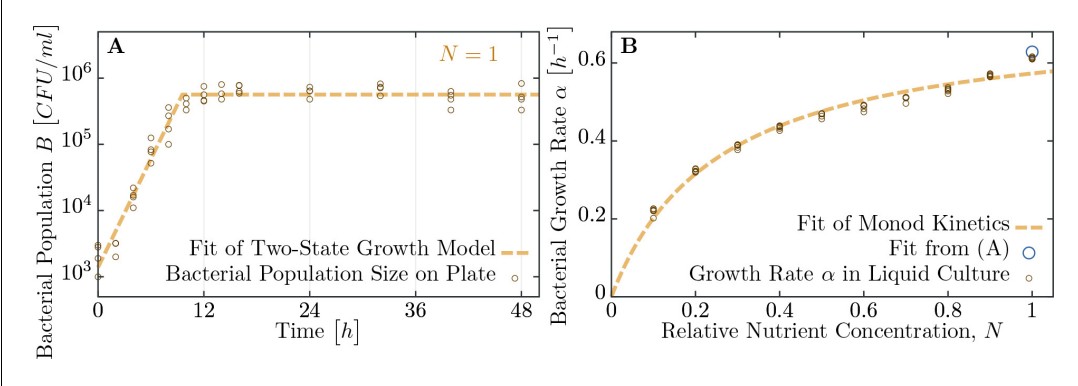

**Figure 4.** Measuring bacterial growth without phage. (**A**) Trajectory of population size on agar plates over time. For modeling, we assume two states of growth (dashed brown curve): first, the bacterial population grows exponential until the time $T_{\text{depl}}$, when nutrients are depleted. From this time on, growth rate is assumed to be zero and the population saturates at a maximal size $B^{\text{final}}$. Experimental observations fit this proposed growth curve to a very good extent. After all, half of all nutrients are used up in the last generation indicating that the switch between growth and no-growth should be fast. (**B**) Growth rates of bacteria in diluted medium follow closely Monod's empirical law, given by expression *Equation 9*. Fit parameters are found to be $\alpha_{\text{max}} \approx 0.720\,h^{-1}$ and $K_c \approx 0.257$ (with the latter in dimensionless units as dilution of LB medium), see also *Table 1*. The data presented in this figure can be found in *Figure 4—source data 1*.

DOI: https://doi.org/10.7554/eLife.32035.008

The following source data is available for figure 4:

**Source data 1.** Bacterial growth on soft agar plates (tab *Figure 4A*) and bacterial growth in LB medium of various dilutions (tab *Figure 4B*).
DOI: https://doi.org/10.7554/eLife.32035.009

The results presented in this and the previous section would allow us to use *Equation 1* to infer a value for $R_0$ from the observed threshold between surviving and collapsing bacterial populations, *Figures 3,5*. We also observe that herd immunity is strongly influenced by spatial organization of the population, *Figure 6*. How the exact value of $H$ (and subsequently the 'classical' epidemiological parameter $R_0$) is affected by various factors such as bacterial growth rate, phage burst size and latent period is, however, difficult to resolve experimentally. Similarly, quantification of the effect of spatial structure is hardly achievable solely by experimental investigation. In order to disentangle the roles of all the factors mentioned above, we proceed with development and analysis of a mathematical model of the experimental system.

## Modelling bacterial herd immunity

We developed a model of phage growth that takes several physiological processes into account: bacterial growth during the experiment, bacterial mortality due to phage infection, and phage mortality due to bacterial immunity. Furthermore, we use the previously reported observation that phage burst size $\beta$ and latent period $\lambda$ depend strongly on the bacterial growth rate $\alpha$ (see *Table 1*).

The main processes in our model system can be defined by the following set of reactions,

$$B_{\text{s}} + yN \xrightarrow{\alpha} 2B_{\text{s}}\,, \tag{2a}$$

$$B_{\text{r}} + yN \xrightarrow{\alpha} 2B_{\text{r}}\,, \tag{2b}$$

$$B_{\text{s}} + P \xrightarrow{A} (B_{\text{s}}P) \xrightarrow{1/\lambda} \beta P\,, \tag{2c}$$

$$B_{\text{r}} + P \xrightarrow{A} (B_{\text{r}}P) \begin{cases} \xrightarrow{\text{fast}} & B_{\text{r}}\,, \\ \xrightarrow{\text{slow}} & \beta P\,. \end{cases} \tag{2d}$$

Susceptible ($B_{\text{s}}$) and resistant ($B_{\text{r}}$) cells grow at a rate $\alpha$ (no significant difference in growth rate between strains, $\alpha(B_{\text{s}}) = 0.551 \pm 0.045\,h^{-1}$, $\alpha(B_{\text{r}}) = 0.535 \pm 0.023\,h^{-1}$, $t_{70} = 1.867, p = 0.066$), *Equation 2*, by using an amount $y$ of the nutrients $N$. Phage infection first involves adsorption to host cells, *Equation 2c* and *Equation 2d*, with the adsorption term $A$ specified below. Infected susceptible bacteria produce on average $\beta$ phage with a rate inversely proportional to the average latency $\lambda$. In contrast, resistant bacteria either survive by restricting phage DNA via their CRISPR-Cas immune system

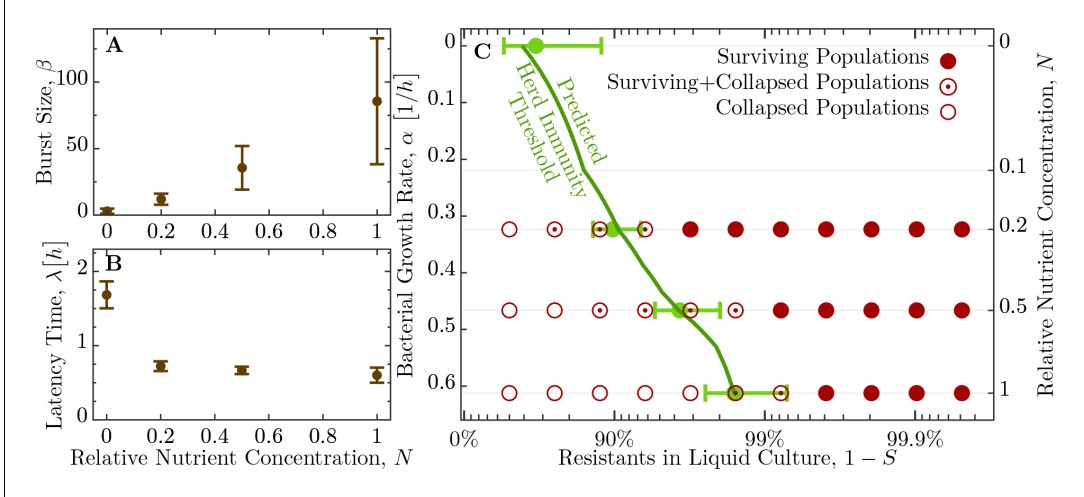

**Figure 5.** Herd immunity threshold in liquid culture as a function of bacterial growth. (A) Phage burst size (β) change as a function of nutrient concentrations. (B) Latent period (λ) increase across the range of nutrient concentrations. Values for β and λ are given in *Table 2*. (C) Population survival analysis upon phage challenge as a function of the fraction of resistant cells and the intrinsic growth rate (nutrient availability, *N*). Bacteria survive the phage infection (full circles), collapse (empty circles), or exhibit both outcomes (circled dots) in the 16 to 18 replicates, done in 3 independent batches. Light green errorbars at investigated dilutions of LB show the expected value and its standard deviation of $H(\alpha)$, *Equation 5*, with standard error propagation of the measured β, λ and α. In order to interpolate herd immunity to dilutions *N* not probed in experiments (dark green line), we use a second order polynomial in *N* to fit the data for both β/λ and β, which excellently matches the average measurements (a naive linear fit displays non-negligible deviations and non-sensical negative values). In addition, the dependence $\alpha = \alpha(N)$ is obtained by numerically inverting the Monod growth rate dependence, see *Equation 9*. The data presented in this figure can be found in *Figure 5—source data 1* and *Figure 5—source data 2*.

DOI: https://doi.org/10.7554/eLife.32035.010

The following source data is available for figure 5:

**Source data 1.** *Figure 5A B* source data: Phage burst sizes and latent period in different dilutions of the growth medium.

DOI: https://doi.org/10.7554/eLife.32035.011

**Source data 2.** *Figure 5C*: Fraction of surviving populations in different dilutions of the growth medium.

DOI: https://doi.org/10.7554/eLife.32035.012

or – less likely – succumb to the phage infection. However, when the MOI is large even resistant cells become susceptible to lysis resulting in the release of phage progeny (see *Figure 2*) (*Westra et al., 2015*; *Chabas et al., 2016*).

In our system, bacteria eventually deplete the available nutrients, $N(t > T_{\text{depl}}) = 0$, resulting in the cessation of growth. This decline in bacterial growth affects phage growth – latency increases and burst size decreases, such that phage reproduction declines dramatically (see *Table 2*). We define the critical time point at which cells transition from exponential growth to stationary phase as,

$$T_{\text{depl}} \approx \frac{1}{\alpha}\log\left(\frac{B_\infty}{B_0}\right). \tag{3}$$

Here, $B_0$ and $B_\infty$ are the initial and final bacterial densities, respectively. In the initial exponential growth phase, our estimates from experimental data for growth parameters are $\alpha = 0.63\,h^{-1}$, $\beta = 85.6\,\text{phages/cell}$ and $\lambda = 0.60\,h$, for bacteria and phages, respectively (*Tables 1* and *2*). After time

**Table 1.** Estimated parameters for bacterial growth using Monod kinetics.

Undiluted LB medium ($N = 1$) is assumed to have $15\,mg/ml$ nutrients ($10\,mg/ml$ Tryptone, $5\,mg/ml$ yeast extract). The full dataset is shown in *Figure 4*.

|  | Estimate | Units |
|---|---|---|
| $\alpha_{\text{max}}$ | $0.720 \sim (\pm 0.011)$ | $[h^{-1}]$ |
| $K_c$ | $0.257 \sim (\pm 0.012)$ | Dilution $N$ of LB $[0\ldots1]$ |

DOI: https://doi.org/10.7554/eLife.32035.013

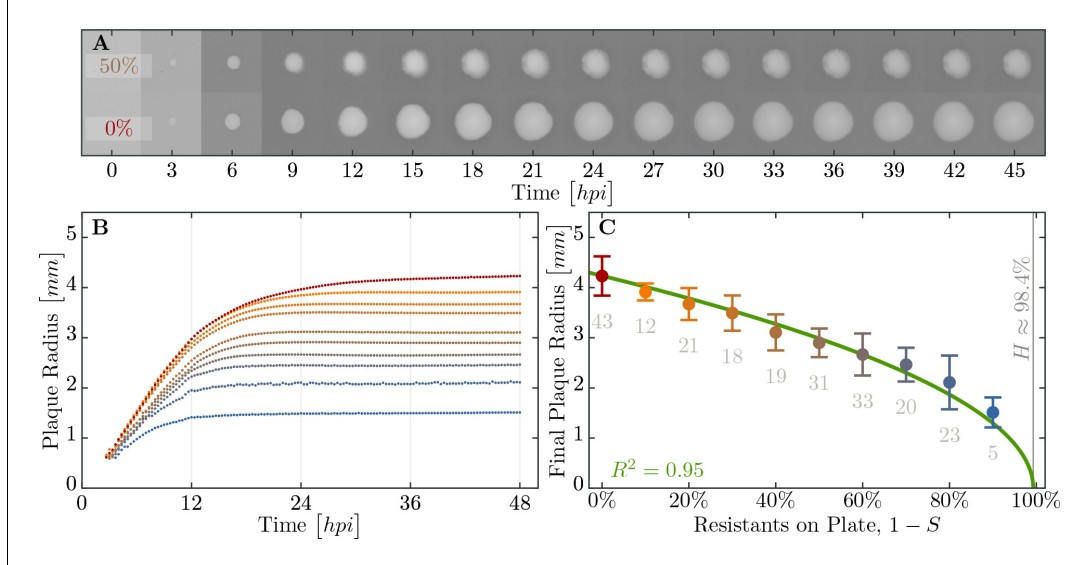

**Figure 6.** Properties of expanding phage epidemics on bacterial lawns. (**A**) Example of plaque morphology and size change over 48 hours for populations with 50% resistant cells (top) and a control with 100% susceptible cells (bottom). (**B**) Mean plaque size area through time. Colors indicate the different fraction of resistant individuals (color coding as in panel C). Note the distinct two phases of plaque growth – initially, phage grow fast with exponentially growing bacteria but slow once the nutrients are depleted ($\approx 10$ hr). The plaque radius is reduced, relative to 100% susceptible population, even when only a small fraction of resistant individuals are in the population. (**C**) Final plaque radius at 48 $hpi$. Green line shows the prediction from the model for the plaque radius $r$. Grey numbers indicate the number of plaques measured. Error bars indicate the standard deviations. The data presented in this figure can be found in **Figure 6—source data 1**.

DOI: https://doi.org/10.7554/eLife.32035.014

The following source data is available for figure 6:

**Source data 1.** Plaque radii for all population compositions and time points.
DOI: https://doi.org/10.7554/eLife.32035.015

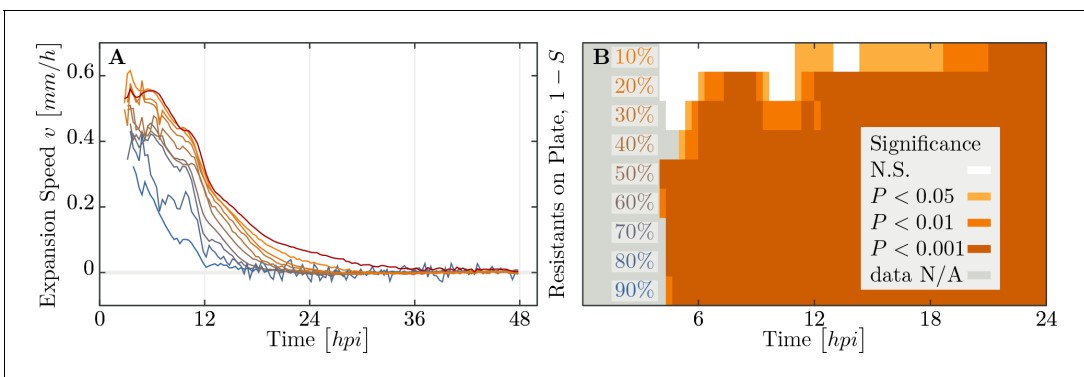

**Figure 7.** Speed of phage epidemic expansion on bacterial lawns. (**A**) Speed of expanding phage epidemics for all population compositions is initially high, before it drops once nutrients are depleted at around $10\,hpi$ (hours post infection). (**B**) Plaque speed significance. Comparing speeds of plaque spread with the 100% susceptible control. Linear regression of a sliding window spanning 4 hours of the radius sizes was calculated for all individual plaques and all compositions of the populations between $t_0$ and $t_{24}$. Slopes of the linear regressions for all compositions of the populations were compared using a two-sided heteroscedastic t-test against the 100% susceptible dataset. The data presented in this figure can be found in **Figure 7—source data 1**.

DOI: https://doi.org/10.7554/eLife.32035.016

The following source data is available for figure 7:

**Source data 1.** Speed of plaque expansion in populations consisting of varying proportions of resistant to susceptible bacteria.
DOI: https://doi.org/10.7554/eLife.32035.017

**Table 2.** Estimated parameters for phage growth.
See also **Figure 5A,B**.

| Medium | Dilution | Latent period | Burst size | Burst size/hour |
|---|---|---|---|---|
| | $N$ | $\lambda\,[min]$ | $\beta$ | $\beta/\lambda\,[h^{-1}]$ |
| LB 0 | 0.0 | 101.1 ($\pm 10.9$) | 3.0 ($\pm 1.9$) | 1.8 ($\pm 1.1$) |
| LB 20 | 0.2 | 43.4 ($\pm 3.9$) | 3.0 ($\pm 1.9$) | 16.6 ($\pm 6.0$) |
| LB 50 | 0.5 | 40.0 ($\pm 3.0$) | 35.6 ($\pm 16.4$) | 53.4 ($\pm 24.9$) |
| LB 100 | 1.0 | 36.1 ($\pm 6.1$) | 85.6 ($\pm 47.3$) | 142.1 ($\pm 82.1$) |

DOI: https://doi.org/10.7554/eLife.32035.018

$T_{\mathrm{depl}}$, bacterial growth rate is set to zero ($\alpha = 0$) and phage growth is reduced to $\beta_{\mathrm{depl}} = 3.0\,\mathrm{phages/cell}$ and $\lambda_{\mathrm{depl}} = 1.69\,h$. Such a two state model – constant growth rate while nutrients are present and no growth after depletion – describes the observed population trajectories in experiments sufficiently well (see **Figure 4**).

## Modelling herd immunity in populations without structure

An important parameter for estimating herd immunity is the fraction $S$ of susceptible bacteria in the population. As a first estimate, a phage infection spreads in well mixed bacterial cultures if $\beta S > 1$, which leads to a continuous chain of infections: the product of burst size $\beta$ of phage particles and the probability $S$ of infecting a susceptible host has to be larger than one. As a first approximation, one could identify $R_0$ with the burst size $\beta$, which is compatible with the observed herd immunity thresholds when inverting **Equation 1**.

However, the growing bacterial population could outgrow the phage population if the former reproduces faster (e.g., in the case of RNA coliphages, van Duin, 1988), which introduces deviations from the simple relation between $R_0$ and $H$ as shown in **Equation 1**. We capture this dynamical effect in a correction to the previous estimate as $\beta S > 1 + \lambda\alpha$ (see Materials and methods): more phages have to be produced for the chain of infections to persist in growing populations. The correction $\frac{\lambda}{1/\alpha}$ is the ratio of generation times of phages over bacteria – usually, such a correction is very small for non-microbial hosts and can be neglected. Ultimately, herd immunity is achieved if the threshold defined by $H = 1 - S_c$ is exceeded, with $S_c$ the critical value in the inequality above. Rearranging, we obtain an expression for the herd immunity threshold

$$H = \frac{\beta - 1 - \lambda\alpha}{\beta} \ . \tag{4}$$

This estimate of $H$ coincides to a very good extent with the population compositions of susceptible and resistant bacteria where we observe the transition from surviving and collapsed populations in experiments (see **Figure 3**). Moreover, simulations presented in the Appendix (section Simulation of recovery rate) show a range in the bacterial population composition with non-monotonic trajectories for $B_s$ and $B_r$ (see **Appendix 1—figure 1B**), which is comparable to the range in composition we find in both outcomes, that is, some surviving and some collapsing populations in experiments. For such parameter choices, stochastic effects could then decide the observed fates of bacteria. As presented above, the herd immunity threshold changes when the bacterial cultures grow in a diluted growth medium. In a set of independent experiments we measured bacterial growth rate $\alpha$, phage burst size $\beta$ and phage latent period $\lambda$ under such conditions (see **Figure 4B** and **Table 2**). From these data we estimated the dependence of the phage burst size on the bacterial growth rate, $\beta(\alpha)$, using a numerical quadratic fit (**Figure 5A**). Similarly, we estimated the dependence of the phage latent period on the bacterial growth rate, $\lambda(\alpha)$ (**Figure 5B**). Using these estimates we calculated the expected growth rate–dependent herd immunity threshold

$$H(\alpha) = \frac{\beta(\alpha) - 1 - \lambda(\alpha)\alpha}{\beta(\alpha)} \ , \tag{5}$$

which gives a very good prediction of the shift in the herd immunity threshold to lower values for

slower growing populations (green line in *Figure 5C*). This verification of our model shows that it correctly captures the dependence of the herd immunity threshold on bacterial and phage growth parameters.

The deviations from the herd immunity threshold depicted by the green area in *Figure 3* and green error bars in *Figure 5C* are derived from uncertainty in measurements in $\beta$, $\lambda$ and $\alpha$. The inherent stochasticity of the adsorption process thus provides additional uncertainty, which is not captured by the depicted error bars. This additional stochasticity can explain wider transition zone in experiments with slower growing populations (dilution 0.5 and 0.2), because the fate of the population is more prone to stochastic effects as the phage replication rate is slower than in a fast growing population. This stochastic effect might be reduced by larger phage inocula. This could, however, also shift the observed transition between collapsing and surviving populations towards higher frequencies of resistant bacteria (and away from the actual herd immunity threshold) as protection by the immune system is less effective with increasing number of phages per cell (see *Figure 2A*).

## Modelling herd immunity in spatially structured populations

The dynamics of phage spread differ between growth in unstructured (e.g., liquid) and structured (e.g., plates) populations. In order to quantify the effect of spatial structure in a population, we extend our model to include a spatial dimension. In structured populations growth is a radial expansion of phages defined by the plaque radius $r$ and the expansion speed $v$, for which several authors have previously derived predictions (*Kaplan et al., 1981*; *Yin and McCaskill, 1992*; *You and Yin, 1999*; *Fort and Méndez, 2002a*; *Ortega-Cejas et al., 2004*; *Abedon and Culler, 2007*; *Mitarai et al., 2016*).

We assume phage movement can be captured by a diffusion process characterized with a diffusion constant $D$, which we estimate in independent experiments as $D = 1.17\,(\pm 0.26) \cdot 10^{-2} \sim mm^2/h$ (see Materials and methods, *Figure 8*). However, we assume that only phages disperse and bacteria are immobile as the rate of bacterial diffusion does not influence the expanding plaque on timescales relevant in the experiment. Adsorption of phages on bacteria is modeled with an adsorption constant $\delta^\star$.

Taking these considerations together, allows to write a reaction-diffusion dynamics for growth of phages $P$ on the growing bacterial population as

$$\partial_t P = D\partial_x^2 P + \delta^\star \left( \beta S - 1 - \lambda \alpha \right) P \,. \tag{6}$$

The first term accounts for the diffusive spread of phages, while the second term describes phage growth. This second term includes the correction $\lambda\alpha$ which arises due reproduction of bacteria, derived in the unstructured liquid case.

The spreading infection will sweep across the bacterial lawn with the following speed

$$v = 2\sqrt{D\delta^\star}\sqrt{\beta S - 1 - \lambda \alpha} \,, \tag{7}$$

which is computed in more details in the Materials and methods. This expression *Equation 7* indicates that the population composition crucially influences the spreading speed at much lower fractions of resistant bacteria than the herd immunity threshold *Equation 4*, where phage expansion stops completely. Consequently, the resulting plaque radius $r$ decays with increasing fractions of resistants and reaches zero at $H$. A prediction for $r$ can be obtained by integrating *Equation 7* over time.

In our (simplified) model, time-dependence of the speed only enters via the fraction of susceptibles $S$, which is assumed to stay at the initial $S_0$ value until it encounters the epidemic wave of phages. Furthermore, we use the experimental observation that plaque expansion ceases upon depletion of nutrients, coinciding with a cessation of bacterial growth. This leads to the approximation $r \approx vT_{\mathrm{depl}}$, with $T_{\mathrm{depl}}$ given by *Equation 3*. Using this expression we estimated the adsorption constant $\delta^\star$ from the growth experiments as it is difficult in practice to measure on plates. The green line in *Figure 6B* is the best fit, yielding the value $\delta^\star = 4.89(\pm 0.19) \cdot 10^{-2}\,\mathrm{bacteria/phage\,}h$ for the adsorption constant.

Our results for spatially structured populations allows us to speculate on a general epidemiological question: If an infection is not stopped by herd immunity in a partially structured population, by

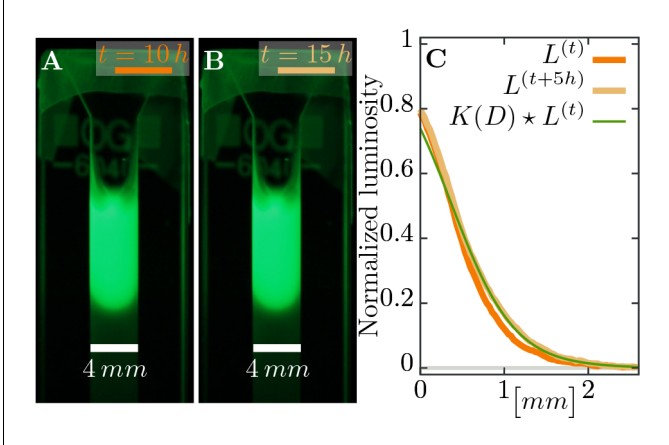

**Figure 8.** Estimating diffusion constant of phages. (A), (B) Phage are slowly expanding on agar which can be observed via their fluorescence. Pictures are taken $5\,h$ apart. (C) The diffusion constant $D$ can be estimated as best-fit parameter in a heat kernel $K(D)$ propagates the fluorescence profile $L^{(t)}$ at time $t$ forward (via a convolution to "smear" out the signal) to the profile $L^{(t+\Delta t)}$ at the next measured time point. The difference between the expected change and the actual profile is quantified as total squared deviation, see *Equation 10*, which we minimize to obtain $D$. Consequently, we can estimate the diffusion constant as $D \approx 1.17 \cdot 10^{-2}\,mm^2/h$. The green line uses this estimated parameter $D$ and shows the change between the profile at $t = 10\,h$ (orange line) and the profile at $t = 15\,h$ (light brown line), assuming diffusive spread of phages. See Materials and methods for more information.
DOI: https://doi.org/10.7554/eLife.32035.019

how much is its spread slowed down? By generalizing *Equation 7* we can derive a relative expansion speed, compared to a fully susceptible population,

$$v_{\mathrm{rel}} = \sqrt{1 - \frac{1-S}{H}}. \tag{8}$$

This expression, *Equation 8*, is devoid of any (explicit) environmental conditions, which are not already contained in the herd immunity threshold $H$ itself. Thus, it could apply to any pathogen-host system. Ultimately, this relative speed approaches zero with a universal exponent of $1/2$, when the fraction of resistant individuals $1-S$ approaches the herd immunity threshold $H$. However, a few caveats exist when using *Equation 8*, as several conditions have to be fulfilled: Obviously, a pathogen is expected not to spread in a population exhibiting complete herd immunity – the relative speed should only hold for populations below the herd immunity threshold. Moreover, if dispersal cannot be described by diffusion, but rather dominated by large jumps (*Hallatschek and Fisher, 2014*), the diffusion approach we used for traveling waves is not applicable, and thus also renders *Equation 8* inadequate.

An increase in the number of long range jumps of phages can be considered as a transition between the two cases we treated here – spatially explicit dynamics on plates and completely mixed populations in liquid culture, respectively. Potential long range jumps of phages can be mediated by host cells moving distances that the phages cannot achieve on their own. In such cases, dispersal of the phages is a convolution of movement of their hosts with their own ability to spread locally. These long range jumps would therefore increase the overall expansion speed and area of the epidemic. We expect that in our setup bacterial motility does not substantially contribute to phage spread because (i) bacteria become motile only in late exponential/early stationary phase (*Amsler et al., 1993*) when phage reproduction drops to very low levels, and (ii) the soft agar concentration used in our experiments ($\approx 0.525\%$) effectively blocks bacterial motility (*Croze et al., 2011*). However, we would not expect that long range jumps change the herd immunity threshold $H(\alpha)$ itself. Spread of pathogens still stops when the fraction of susceptible hosts $S$ is small such that $\beta S < 1 + \lambda\alpha$, and will continue as long as $\beta S > 1 + \lambda\alpha$ is fulfilled.

## Discussion

The spread of a pathogen may be halted or slowed by resistant individuals in a population and thus provide protection to susceptible individuals. This process, known as herd immunity, has been extensively studied in a wide diversity of higher organisms (*Jeltsch et al., 1997*; *Mariner et al., 2012*; *van Boven et al., 2008*; *Meister et al., 2008*; *Konrad et al., 2012*; *Wang et al., 2013*). However, the role of this process has largely been ignored in microbial communities. To delve into this we set out to determine under what conditions, if any, herd immunity might arise during a phage epidemic in bacterial populations as it could have profound implications for the ecology of bacterial communities.

We show that herd immunity can occur in phage-bacterial communities and that it strongly depends on bacterial growth rates and spatial population structure. Average growth rates of bacteria in the wild have been estimated as substantially slower than in the laboratory (generation time is ≈ 7.4 fold longer [*Gibson et al., 2017*]), a condition that we have shown to facilitate herd immunity. Furthermore, bacterial populations in the wild are also highly structured, as bacteria readily form micro-colonies or biofilms (*Hall-Stoodley et al., 2004*) and grow in spatially heterogeneous environments such as soil or the vertebrate gut (*Fierer and Jackson, 2006*), a second condition we have shown to facilitate herd immunity. These suggest that herd immunity may be fairly prevalent in low nutrient communities such as soil and oligotrophic marine environments.

In an evolutionary context, herd immunity may also impact the efficacy of selection as the selective advantage of a resistance allele will diminish as the frequency of the resistant allele in a population approaches the herd immunity threshold, $H$. This has two important implications. First, while complete selective sweeps result in the reduction of genetic diversity at linked loci, herd immunity may lead to only partial selective sweeps in which some diversity is maintained. Second, herd immunity has a potential to generate and maintain polymorphism at immunity loci, as has been shown for genes coding for the major histocompatibility complex (MHC) (*Wills and Green, 1995*). Polymorphism in CRISPR spacer contents have been demonstrated in various bacterial (*Tyson and Banfield, 2008*; *Sun et al., 2016*; *Kuno et al., 2014*) and Archaeal (*Held et al., 2010*) populations and communities (*Pride et al., 2011*; *Zhang et al., 2013*; *Andersson and Banfield, 2008*). While these studies primarily explain polymorphisms in CRISPR spacer content as a result of rapid simultaneous independent acquisition of new spacers, we suggest that observed polymorphisms may result from frequency-dependent selection on resistance loci arising from herd immunity. In such a case, herd immunity is likely to maintain existing polymorphism in CRISPR spacer content in $1 - H$ fraction of the population, unless the current major variant goes to fixation due to drift. However, considering the large population sizes of bacteria, drift is unlikely to have a strong effect, allowing herd immunity to maintain a large fraction of immunity polymorphism.

It has also been suggested that herd immunity might favor coexistence between hosts and their pathogens (*Hamer, 1906*), which can lead to cycling in pathogen incidence and proportions of resistant and susceptible individuals over time, e.g., in measles before the era of vaccination (*Fine, 1993*). This cycling is caused by the birth of susceptible individuals, which, once their proportion exceeds the epidemic threshold $(1 - H)$, lead to recurring epidemics. CRISPR-based immunity is, however, heritable meaning that descendants of resistant bacteria remain resistant. One might speculate that analogous cycling in phage epidemics may occur if immunity is costly. In turn, a computer simulation study of coevolution of Streptococcus thermophilus and its phage found both cycling and stable coexistence of different CRISPR spacer mutants and phage strains (*Childs et al., 2014*). The extent to which herd immunity facilitates maintenance of CRISPR spacer polymorphism and coexistence with phage requires further experimental and theoretical investigation.

We also developed a mathematical model and show how the herd immunity threshold $H$ (*Equation 4*) depends on the phage burst size $\beta$ and latent period $\lambda$, and on the bacterial growth rate $\alpha$. This dependence arises as phages have to outgrow the growing bacterial population, as host and pathogen have similar generation times in our microbial setting. In addition to these parameters, we also describe how the speed $v$ (*Equation 7*) of a phage epidemic in spatially structured populations depends on phage diffusion constant $D$, phage adsorption rate $\delta^*$, and the fraction of resistant and susceptible individuals in the population. All of which are likely to vary in natural populations. We also derived the relative speed of spread for partially resistant populations, as measured relative to a fully susceptible population, and show that it can be parametrized solely with the herd immunity

threshold $H$ (*Equation 8*). This relative speed of the spread of an epidemic should be applicable to any spatially structured host population where the spread of the pathogen can be approximated by diffusion. Both our experiments and the modelling show that even when the fraction of resistant individuals in the population is below the herd immunity threshold the expansion of an epidemic can be substantially slowed, relative to a fully susceptible population.

In conclusion, we have presented an experimental model system and the connected theory that can be usefully applied to both microbial and non-microbial systems. Our theoretical framework can be useful for identifying critical parameters, such as $H$ (and to some extent $R_0$), from the relative speed of an epidemic expansion in partially resistant populations so long as the process of pathogen spread can be approximated by diffusion. This approximation has been shown to be useful in such notable cases as rabies in English foxes (*Murray et al., 1986*), potato late blight (*Scherm, 1996*), foot and mouth disease in feral pigs (*Pech and McIlroy, 1990*), and malaria in humans (*Gaudart et al., 2010*).

# Materials and methods

**Key resources table** Table of key strains, reagents and software used in this study.

| Reagent type (species) or resource | Designation | Source or reference | Identifiers | Additional information |
|---|---|---|---|---|
| gene (Streptococcus pyogenes SF370) | cas9 | National Center for Biotechnology Information | NCBI:NC_002737.2; gene_ID:901176; RRID:SCR_006472 | Gene symbol SPy_1046 |
| strain, strain background (Escherichia coli) | E. coli K12 MG1655 | Own collection | NA | |
| strain, strain background (Bacteriophage T7) | E. coli bacteriophage T7 | ATCC Collection | ATCC:BAA-1025-B2; RRID:SCR_001672 | |
| recombinant DNA reagent | pCas9 | Addgene Vector Database | Addgene:42876; RRID:SCR_005907 | pCas9 plasmid was a gift from Luciano Marraffini |
| recombinant DNA reagent | pCas9T7resistant | this paper | NA | Plasmid derived from pCas9 |
| commercial assay or kit | PureYield Plasmid Miniprep System | Promega | Promega:A1223; RRID:SCR_006724 | |
| chemical compound, drug | Chloramphenicol | Sigma-Aldrich | Sigma-Aldrich:C0378-5G; RRID:SCR_008988 | |
| software, algorithm | PerkinElmer Volocity v6.3 | | RRID:SCR_002668 | Volocity 3D Image Analysis Software |
| software, algorithm | Fiji v1.0 | doi: 10.1038/nmeth.2019 | RRID:SCR_002285 | Image processing package of ImageJ |
| software, algorithm | RStudio 1.0.153 | | RRID:SCR_000432 | Software for the R statistical computing |
| software, algorithm | Python 3.6.3 | | RRID:SCR_008394 | Python programming language |
| software, algorithm | Model source code | doi: 10.5281/zenodo.1038582 | RRID:SCR_004129 | Zenodo repository |

## Experimental methods

### Engineering resistance

Oligonucleotides AAACTTCGGGAAGCACTTGTGGAAG and AAAACTTCCACAAGTGCTTCCCGAA were ordered from Sigma-Aldrich, annealed and ligated into pCas9 plasmid (pCas9 was a gift from Luciano Marraffini, Addgene plasmid #42876) carrying a Streptococcus pyogenes truncated CRISPR type II system and conferring chloramphenicol resistance. For the detailed protocol see (*Jiang et al., 2013*). The oligonucleotides were chosen so that the CRISPR system targets an overlap of phage T7 genes 4A and 4B. Therefore, the CRISPR system allows the gene 0.7, coding for a protein which inhibits the RNA polymerase of the cell, to be expressed before the T7 DNA gets cleaved (*García and Molineux, 1995*). The subsequent growth of the cells is halted and phage replication is inhibited. The plasmid was electroporated into Escherichia coli K12 MG1655 (F- lambda- ilvG- rfb-50 rph-1). The T7 wildtype phage was used in all experiments.

## Efficiency of the CRISPR-Cas system

Efficiency of the engineered CRISPR-Cas system was tested using the following protocol: Overnight cultures grown in LB containing 25 $\mu g/ml$ chloramphenicol were diluted 1 in 10 in the same medium, cells were infected with the T7 phage (MOI 10, 100, and 1000) and incubated for 15 min in 30°C. Cells were spun down for 2 min in room temperature at 21130g. Supernatant was discarded and the cell pellet was resuspended in 950 $\mu l$ of 1X Tris-HCl buffer containing 0.4% ($\approx 227 \mu M$) ascorbic acid pre-warmed to 43°C and incubated in this temperature for 3 min to deactivate free phage particles (*Murata and Kitagawa, 1973*). Cultures were serially diluted and plated using standard plaque assay protocol on a bacterial lawn of susceptible cells to detect bursting infected cells. The supernatant was tested for free phage particles, which were not detected in the corresponding dilutions used for plaque counting. Each experiment was replicated three or four times (MOI 10 three times, MOI 100 four times and MOI 1000 three times) while samplings from each treatment were performed in quadruplicates. The probability that a resistant cell bursts was calculated as a ratio of bursting resistant to bursting susceptible cells for each experiment (means of corresponding quadruplicates). All LB agar plates and soft agar used throughout this study was supplemented with 25 $\mu g/ml$ chloramphenicol. These CRISPR-Cas system efficiencies at different MOIs were tested if they are statistically different from each other using two-tailed unequal variances t-test at 0.05 confidence level using RStudio (*R Core Team, 2013*).

## Determining the mean number of phages per cell

The cultures that were plated using standard plaque assays in the "Efficiency of the CRISPR-Cas system" experiment were also plated on LB agar plates containing 25 $\mu g/ml$ chloramphenicol to determine the number of surviving CFUs. The numbers of bursting and surviving susceptible cells were subsequently used to determine the actual mean number of adsorbed phages per cell. The fraction of susceptible cells surviving the phage challenge experiment was assumed to correspond to the Poisson probability that a cell does not encounter any phage, which was than used to determine the mean of the Poisson distribution, which corresponds to the mean number of phages per cell.

## Herd immunity in a liquid culture

Herd immunity in a liquid culture was tested in 200 $\mu l$ of LB broth supplemented with 25 $\mu g/ml$ chloramphenicol in Nunclon flat bottom 96 well plate in a Bio-Tek Synergy H1 Plate reader. Bacterial cultures were diluted 1 in 1000 and mixed in the following ratios of resistant to susceptible cells: 50:50, 75:25, 87.5:12.5, 93.75:6.25, 96.88:3.13, 98.44:1.56, 99.22:0.78, 99.61:0.39, 99.8:0.2, 99.9:0.1, 99.95:0.05, 100:0 %. T7 phage was added at a multiplicity of infection (MOI) of $\approx 10^{-4}$ ($\approx 50$ plaque forming units (*pfu*) per culture) to resemble an epidemic initiated by the burst size from one infected cell and the cultures were monitored at an optical density 600 nm for 18 hours post inoculation (*hpi*). Each population composition was replicated 16 times. Herd immunity in diluted LB was measured in LB broth mixed with 1X M9 salts in ratios 1:1 (50% LB) and 1:4 (20% LB) using the same protocol as for 100% LB broth. Each population composition was replicated 18 times.

## Time-lapse imaging of plaque growth

Soft LB agar (0.7%) containing 25 $\mu g/ml$ chloramphenicol was melted and 3 $ml$ were poured into glass test tubes heated to 43°C in a heating block. After the temperature equilibrated, 0.9 $ml$ of a bacterial culture consisting of resistant and susceptible cells (ratios 10% – 100% of susceptible cells, 10% increments) were diluted 1 in 10 and added to the tubes. Then, 100 $\mu l$ of bacteriophage stock, diluted such that there would be $\approx 10$ plaques per plate, was added to the solution. Tubes were vortexed thoroughly and poured as an overlay on LB agar plates containing 25 $\mu g/ml$ chloramphenicol. The plates were placed on scanners (Epson Perfection V600 Photo Scanner) and scanned every 20 minutes in "Wide Transparency mode" for 48 hours in 30°C. A total of 3 scanners were employed with a total of 12 plates, plus a no phage control plate and 100% resistant control outside the scanners (see Appendix figure 3). No plaques were detected in the 100% resistant controls. Time-lapse images were used to calculate the increase of individual plaque areas using image analysis software PerkinElmer Volocity v6.3 and Fiji v1.0 (*Schindelin et al., 2012*).

## Bacterial growth on soft agar

Growth rate of susceptible bacteria in soft LB agar (0.7%) was measured by sampling from a petri dish with a soft agar overlay with bacteria prepared in the same way as the plaque assays except an absence of the phage. Sampling was performed in spatially randomized quadruplicates at the beginning of the experiment and subsequently after 2, 4, 6, 8, 10, 12, 14, 16, 24, 32, 40, and 48 hours using sterile glass Pasteur pipettes (Fisherbrand art.no.: FB50251). Samples were blown out from the Pasteur pipette using an Accu-jet pro pipettor into 1 $ml$ of M9 buffer pre-warmed to $43°C$, vortexed for 15 seconds and incubated for 10 minutes in $43°C$ with two more vortexing steps after 5 and 10 minutes of incubation. Samples were serially diluted and plated on LB agar plates containing 25 $\mu g/ml$ chloramphenicol. How bacterial densities change over time, measured as $CFU/ml$, is shown in **Figure 4A**.

## Bacterial growth rates in liquid culture

Nutrient-dependent growth rate of susceptible bacteria was measured in Nunclon flat bottom 96 well plate in Bio-Tek Synergy H1 Plate reader in $30°C$. Overnight LB cultures were diluted 1:200 in media consisting of LB broth mixed with 1X M9 salts in ratios 10:90, 20:80, 30:70, 40:60, 50:50, 60:40, 70:30, 80:20, 90:10 and 100:0. Final volume was 200 $\mu l$. Optical density at 600 nm was measured every 10 min. Every treatment was replicated eight times. Natural logarithm of the optical density values was calculated to determine the growth rate using a maximal slope of a linear regression of a sliding window spanning 90 min.

The resulting growth rates for various nutrient concentrations fit well with Monod's growth kinetics,

$$\alpha = \alpha_{\max}\frac{N}{K_c + N} \ . \tag{9}$$

Results for the two fitting parameters, $\alpha_{\max}$ and $K_c$, are listed in **Table 1**. The whole dataset, including the fit, is displayed in **Figure 4B**.

Test for a difference in growth rates of resistant and susceptible bacteria was done in LB broth in the same manner as nutrient-dependent growth rate measurements. Two-sample t-test was performed on acquired growth rate data at 0.05 confidence level using RStudio (**R Core Team, 2013**).

All growth media used in growth rate measurements were supplemented with 25 $\mu g/ml$ chloramphenicol.

## Phage burst sizes

Phage burst sizes in bacteria growing at different growth rates were measured by one-step phage growth experiments in LB mixed with 1X M9 salts in the following ratios 0:100, 20:80, 50:50 and 100:0. The burst sizes were calculated as the ratio of average number of plaques before burst to average number of plaques after burst. Consecutive samplings before and after burst were used for the calculation if they were not significantly different from each other (two sided t-test, $p > 0.05$). All experiments were performed in triplicates.

## Phage latent periods

Phage latent periods were determined from the phage burst size experiments as the time interval between the first and the last significantly different consecutive sampling between those used for phage burst size calculations.

## Speed of phage expansion

The speed of the phage expansion was measured as difference in radii of plaques over time. Statistical tests allowed to infer that the reduction of expansion speed is significant already for small deviations from the $100\%$ susceptible control experiment, as described and shown in **Figure 7**.

## Phage diffusion in soft agar

Soft M9 salts soft agar (0.5%) was supplemented with SYBR safe staining (final conc. 1%) and poured into glass cuvettes (VWR type 6040-OG) to fill $\sim 2\,cm$ of the cuvette height. After soft agar

solidification, the same stained soft agar was supplemented with T7 phage particles to a final concentration $10^{11}$ pfu/ml and poured on top of the agar without phages. The cuvettes were monitored in $30°C$ every hour for 40 hours at the SYBR safe emission spectrum peak wave length $524\,nm$ illuminated with the SYBR safe excitation spectrum peak wave length $509\,nm$. The diffusion constant was estimated as the best fit parameter for the spread of fluorescent phages through the soft agar over time.

First we computed the luminosity $L_i$ of fluorescence (a gray-scale value defined as $L = 0.2126R + 0.7152G + 0.0722B$ from the RGB image) as average over the width of the cuvette for pixel row $i$, and corrected the profiles of luminosity $L_i$ by subtracting the background value. This background value was estimated as a linear fit at the end of the profile without phages, where only the gray value of the agar was measured. Moreover, in our setup luminosity seems to saturate at values above $\sim 0.4$ where it does not have a simple linear dependence on fluorescence anymore: diffusion would lead to a decrease of the signal behind the inflection point of the profile and increase after the inflection point, but images only show increasing profiles – the bulk density does not decay. Thus, any estimate should only take the part of the profile that is below the threshold value of $0.4$ into account (see *Figure 8*).

The diffusion constant $D$ itself was estimated as the minimal value of the total squared deviation of the convoluted profile $L^{(t)}$ (at time $t$) with a heat kernel $K(D)$ compared to the profile $L^{(t+1)}$ at time $t + 1$,

$$D = \left\langle \min_D \sum_i \left( \left( \sum_j \frac{e^{-(i-j)^2/4D}}{\sqrt{4\pi D}} L_j^{(t)} \right) - L_i^{(t+1)} \right)^2 \right\rangle . \tag{10}$$

Such a convolution with the heat kernel $K_{ij}(D) = (4\pi D)^{-1/2} \exp\left(-(i-j)^2/4D\right)$ assumes that the only change in the profile is due to diffusion for a time span of length 1 with $i$ and $j$ indices of pixels. Thus, expression *Equation 10* estimates the diffusion constant in units of $\mathrm{pixel}^2/\mathrm{frame}$, where frame is the time difference between two images. Several estimates are averaged over different snapshots in the whole experiment that spans $40\,h$ in intervals of $1\,h$ each.

The final estimate in appropriate units is

$$D \approx 1.17 \, (\pm 0.26) \cdot 10^{-2} \sim mm^2/h \,, \tag{11}$$

which is in agreement with previous measures of phage diffusion (*Stent and Wollman, 1952*; *Bayer and DeBlois, 1974*; *Briandet et al., 2008*).

## Modelling
### Phage growth
In the main text we stated that relevant processes for phages growing on bacteria are given by the set of reactions *Equation 2*. In the following, we will analyze an extended version of our model, which takes all these processes into account. We try to justify our approximations and explain the reasoning behind leaving parts of the full model out. While reactions for single bacteria or phages are inherently stochastic in nature, we assume that the involved numbers are large enough such that the dynamics can be described with deterministic differential equations for the populations. Furthermore, reaction rates are identified with the inverse of the average time scale of the process. Thus, the full model is given by the coupled differential equations,

$$\partial_t B_s = \alpha B_s - A[B_s, P | B_s, B_r] \,, \tag{12a}$$

$$\partial_t B_r = \alpha B_r - A[B_r, P | B_s, B_r] + \rho I_r \,, \tag{12b}$$

$$\partial_t I_s = A[B_s, P | B_s, B_r] - (1/\lambda) I_s \,, \tag{12c}$$

$$\partial_t I_r = A[B_r, P | B_s, B_r] - (1/\lambda) I_r - \rho I_r \,, \tag{12d}$$

$$\partial_t P = (\beta/\lambda)(I_s + I_r) - \sum_{i \in \{s,r\}} A[B_i, P | B_s, B_r] - \sum_{i \in \{s,r\}} A[I_i, P | I_s, I_r] \,, \tag{12e}$$

$$\partial_t N = -\alpha/y (B_s + B_r) \,. \tag{12f}$$

Both bacterial populations $B_i$, $i \in \{s, r\}$, grow with rate $\alpha$ and decay via adsorption of phages

$A[B_i, P|B_{\mathrm{s}}, B_{\mathrm{r}}]$, an expression that is specified below. Infected populations $I_i$ gain numbers by adsorption and decrease via bursting. Resistant bacteria also can recover from their infected state with a recovery rate $\rho$. Phages grow by bursting cells, and lose numbers by adsorption to the various bacterial populations. Moreover, explicit dynamics for nutrients is considered, which are drained by each grown cell inversely proportional to the yield $Y$, the conversion factor between nutrient concentration and cell numbers. Essentially, this last equation acts as a timer, when we switch from abundant resources to the depleted state: all growth parameters change significantly upon nutrient depletion. Nevertheless, despite the possible deviations, we assume depletion time is given by the simple estimate *Equation 3* and only treat the two possible states of abundant and depleted nutrients.

Adsorption of phages, given by the term $A[B_i, P|B_{\mathrm{s}}, B_{\mathrm{r}}]$, can be influenced by the whole distribution of populations within the culture. In liquid medium, a common assumption is that this term is proportional to the concentrations of both the phages and cells (*Weitz, 2016*),

$$A[B_i, P|B_{\mathrm{s}}, B_{\mathrm{r}}] = \delta B_i P, \tag{13}$$

with an adsorption constant $\delta$. This expression assumes constant mixing of the population and relatively short contact times between phages and bacteria. In general, this system of equations is akin to Lotka-Volterra dynamics, which has been analyzed in great detail, eg. (*Hofbauer and Sigmund, 1998*; *Nowak, 2006*).

For our ensuing analysis, we neglect the population of infected resistant bacteria $I_{\mathrm{r}}$. Upon examining *Equation 12* we find that most cells to leave their infected state by reducing phage DNA via CRISPR/Cas instead of bursting if $\rho \gg 1/\lambda$. If furthermore $\rho \gg \delta P$, which is true at least in the initial stages of the experiment, essentially all infected resistant bacteria immediately recover from a phage infection. Consequently, with both conditions, the resistant infected bacteria tend to vanish, $I_{\mathrm{r}} \to 0$, and their dynamics can be neglected. Only in the Appendix (section Simulation of recovery rate) we release this assumption to explicitly cover the full dynamics of *Equation 12* in simulations to estimate values for $\rho$.

## Exponentially growing bacteria lead to double exponential phage growth

For convenience, we transform the populations to the total bacterial density $B = B_{\mathrm{s}} + B_{\mathrm{r}}$ and introduce the fraction of susceptible cells $S = B_{\mathrm{s}}/B$. The crucial assumption for the remainder of this section is that phages burst immediately after infection, $\lambda = 0$, such that we can ignore all infected populations. While not a very biological condition, it allows to analyze the model in more detail. Using these simplifications, we obtain

$$\partial_t B = (\alpha - \delta S P)B, \tag{14a}$$
$$\partial_t S = -S(1 - S)\delta P, \tag{14b}$$
$$\partial_t P = (\beta S - 1)\delta B P. \tag{14c}$$

If we assume that in initial stages of phage growth the number of phages is small, ie. $\delta P \ll \alpha \sim \mathcal{O}(1\,h^{-1})$, the dynamics of bacteria and the fraction of susceptibles simplify to $\partial_t B = \alpha B$ and $\partial_t S = 0$. Note that this term $\delta P$ also occurs in the linear phage dynamics, where it cannot be neglected. In this instance, we need to view $\delta B$ as a coefficient, which is likely much larger initially. This set of simplified equations can be solved in closed form,

$$S(t) = S_0, \tag{15a}$$
$$B(t) = B_0 \exp(\alpha t), \tag{15b}$$
$$P(t) = P_0 \exp\big((S_0\beta - 1)\delta B_0(\exp(\alpha t) - 1)/\alpha\big). \tag{15c}$$

The structure of phage dynamics is particularly important here – it exhibits a double-exponential dependence on time $t$, which is a very fast, almost explosive, growth. Such double-exponential growth leads to very large population sizes within a short amount of time (but after an extended initial delay). This general behavior of the solution is independent of the actual growth rate of phages, which only has to be positive. Thus, inspecting the exponent in *Equation 15* yields the condition

$$\beta S_0 > 1 \tag{16}$$

for phage growth to be positive. Incidentally, relation *Equation 16* is the naive estimate for the number of successful additional infections arising from a single burst. The double exponential time-dependence is central for our arguing that the dynamics can be described by threshold phenomena, given by conditions like *Equation 16*: Usually, phages are negligible in the dynamics until they grow fast enough to large enough size, such that it is too late for the bacterial population to deal with the overwhelming phage population.

An important question in the context of these solutions is whether nutrients run out before this double-exponential growth of phages occurs. Hence, we compute the time $T_\delta$ defined as when phages reach a population of $P(T_\delta) = 1/\delta$ assuming phages grow as *Equation 15* until then. After $T_\delta$ the assumptions that allowed to obtain *Equation 15* are not valid anymore. Inverting *Equation 15* for time leads to

$$T_\delta = \frac{1}{\alpha} \log\left(1 + \frac{\alpha \log(1/\delta P_0)}{(\beta S_0 - 1)\delta B_0}\right). \tag{17}$$

Subsequently, we can compare this estimate $T_\delta$ to the depletion time $T_{\text{depl}} = (1/\alpha)\log(B_\infty/B_0)$. When rearranging the inequality $T_{\text{depl}} > T_\delta$ in terms of the (initial) fraction of susceptibles $S_0$, we obtain

$$\beta S_0 > 1 + \frac{\alpha \log(1/\delta P_0)}{\delta(B_\infty - B_0)}. \tag{18}$$

This expression *Equation 18* is a condition for phages to reach "large" population sizes before nutrients are depleted by bacteria. The final population density $B_\infty$ usually fulfills $\delta B_\infty/\alpha \gg 1$, such that the correction given by the second term of *Equation 18* can be considered small. Thus, if phages grow ($\beta S_0 > 1$), they also grow very fast with a double-exponential time-dependence and reach a considerably large population size before bacteria stop multiplying (for almost all parameter values).

## Extending analysis to finite burst times

The analysis above only treated the case $\lambda \to 0$. However, we reported that the latency time $\lambda$ increases significantly when bacterial growth rate $\alpha$ declines, see *Table 2*. Considering finite latency times entails dealing with an infected bacterial population $I$. (However, we identify $I \equiv I_s$ and set $I_r = 0$.)

To this end, note that we can rearrange *Equation 12* to $(1 + \lambda \partial_t)I = \lambda \delta SBP$ using the adsorption model in *Equation 13*. Hence, we can use the differential operator $(1 + \lambda \partial_t)$ and apply it directly to *Equation 12e* to reduce the dependence on $I$ in this equation at the cost of introducing higher order derivatives. In particular, we obtain

$$\lambda \partial_t^2 P + (1 + \lambda \delta B)\partial_t P + \delta B(\beta S - 1 - \lambda \alpha)P = 0, \tag{19}$$

where we also inserted $\partial_t B \approx \alpha B$ in the last term, as we aim again for a solution at initial times where $\delta P \ll \alpha$. The effects of the limit $\lambda \to 0$ are directly observable – no terms are undefined in this limit. In particular, we find that equation *Equation 19* and $\lambda = 0$ lead directly to the dynamics of phages we just analyzed above, obtaining solution *Equation 15*.

In principle, *Equation 19* is a hyperbolic reaction-diffusion-equation, which is known to occur upon transformation (or approximation) of time-delayed differential equations (*Fort and Méndez, 2002b*). For initial times we can use the solutions $B(t) = B_0 \exp(\alpha t)$ and $S(t) = S_0$. To proceed, we introduce the auxiliary variable

$$z(t) = -\delta B_0 \exp(\alpha t)/\alpha, \tag{20}$$

and assume $P(z)$ as a function of this new variable $z$. We need to transform the differential operators of time derivatives, and obtain $\partial_t = \frac{\partial z(t)}{\partial t}\partial_z = \alpha z \partial_z$ and $\partial_t^2 = (\alpha z \partial_z)(\alpha z \partial_z) = \alpha^2(z\partial_z + z^2\partial_z^2)$. Inserting these expressions in *Equation 19* and multiplying the whole equation with $(\alpha^2 \lambda z)^{-1}$ yields the dynamics for phages,

$$0 = z\partial_z^2 P(z) + (b - z)\partial_z P(z) - aP(z), \tag{21}$$

where the two extant constants are $a = 1 - (\beta S_0 - 1)/(\lambda\alpha)$ and $b = 1 + 1/(\lambda\alpha)$. *Equation 21* is called 'Kummer's equation' with confluent hypergeometric functions $_1F_1$ as solutions (*Abramowitz and Stegun, 1964*, pg. 504),

$$P(z) = A\,_1F_1\big(a, b; z\big) + B z^{1-b}\,_1F_1\big(a - b + 1, 2 - b; z\big)\,. \tag{22}$$

The two integration constants $A$ and $B$ can be determined via the initial conditions $P(t = 0) = P_0$ and $(\partial_t P)(t = 0) = -\delta B_0 P_0$. Using these conditions, the shape of the solution is again similar to before with $\lambda = 0$ (double exponential time-dependence), although $\lambda > 0$ introduces some skew. The most important aspect of this solution *Equation 22* is to compute the parameter combination where it switches from a decreasing to increasing function over time. A careful analysis reveals that at the parameter value $a = 0$ the behavior of the solution changes. Consequently, we find the condition for growing phage populations,

$$\beta S_0 > 1 + \lambda\alpha\,, \tag{23}$$

which is a non-trivial extension including finite latency times $\lambda$.

Note, however, that this relation *Equation 23* does not indicate a correction to the general epidemiological parameter $R_0$, which can be identified with $\beta$ in our model. Rather, it shows that a growing bacterial population requires the phage population to grow even faster for a continuous chain of infections in an epidemic. The term $\lambda\alpha$ denotes the ratio of generation times of pathogen over host, which in most cases is small and negligible compared to 1. For bacteria and phages, however, which have similar generation times, such a correction is needed to describe the effects of growing host population sizes. In contrast, many other epidemiological models assume the host population size constant and only pathogens are increasing (or decreasing) in number.

While our result *Equation 23* suggest that it also should hold in the limit $\alpha \to 0$, it might not necessarily be so. This specific limit is actually quite important for the time when nutrients are depleted in the experiments. However, at several instances in the calculations above we implied a positive $\alpha > 0$. The most important of these is the transformation to $z(t) = -\delta B(t)/\alpha$, which actually exhibits two problems: dividing by $\alpha$ should not be allowed and $B(t)$ is essentially constant and cannot serve as a variable in a differential equation. We also neglected the second term in $\partial_t B = (\alpha - \delta SP)B$ throughout our calculation. For $\alpha = 0$ this second term is dominant in bacterial dynamics and would generate non-linear phage dynamics if inserted for $\partial_t B$ right before stating *Equation 19*. However, we expect that albeit the process will run very slow, and might not be measurable in experiments, the simple condition $\beta S_0 > 1$ could indicate phage expansion and bacterial decay.

## Growth of phages on plated bacterial lawn

Spatial modelling of phage expansion has produced several predictions for how plaque radius $r$ and expansion speed $v$ are influenced by experimentally adjustable parameters (*Kaplan et al., 1981*; *Yin and McCaskill, 1992*; *You and Yin, 1999*; *Fort and Méndez, 2002a*; *Ortega-Cejas et al., 2004*; *Abedon and Culler, 2007*; *Mitarai et al., 2016*). Here, we try to use a minimal model to estimate these two observables, based on the considerations of previous sections.

One of the main complications arises from the fact that all densities in *Equation 12* have a spatial dimension in addition to their time dependence, $B_i = B_i(\vec{x}, t)$, $i \in \{\mathrm{s}, \mathrm{r}\}$. As explained in the main text we only consider phage diffusion, heterogeneities in all other densities are generated only by phage growth. The additional spatial dimension imposes a particular contact network between phages and bacteria, which are not entirely random encounters anymore: One can expect that the size of the bacterial neighborhood $\hat{B}$ phages are able to explore is only slightly determined by the actual density $B$, and can be assumed constant for most of the experiment, $\hat{B}(B) \approx const$. Consequently, the adsorption term can be written in the following way,

$$A\big[B_i, P | B_\mathrm{s}, B_\mathrm{r}\big] = \delta^\star \frac{B_i}{B_\mathrm{s} + B_\mathrm{r}} P\,, \quad i \in \{\mathrm{s}, \mathrm{r}\}\,, \tag{24}$$

which only depends on the relative frequencies of bacterial strains. The adsorption constant $\delta^\star$ is both the rate of adsorption and inter-host transit time as determined by the diffusion constant $D$. Thus, one can expect the formal dependence $\delta^\star = \delta^\star\big(D, \hat{B}(B)\big)$. For our particular experimental setup,

however, $\delta^\star$ will be treated as a constant. This adsorption term *Equation 24* leads to the dynamics of phages

$$\partial_t P = D\nabla^2 P + G[P,S]\,,\tag{25}$$

where we collected all contributions to phage growth in $G[P,S]$ and added the spatial diffusion term $D\nabla^2 P$. For simplicity, we consider only expansion in a single dimension ($\nabla^2 \equiv \partial_x^2$), which has been found to coincide well with the dynamics of plaque growth (*Yin and McCaskill, 1992*). The growth term for phages is then defined as,

$$G[P,S] = \delta^\star\big(S\beta - 1 - \lambda\alpha\big)P\,,\tag{26}$$

where we also consider the correction $\lambda\alpha$ obtained from the analysis in liquid culture. Due to the different absorption dynamics on plates, however, this correction might be a slight overestimate of the actual term that accounts for bacterial growth. Reaction-diffusion equations similar to *Equation 25* have been first analyzed about 80 years ago (*Fisher, 1937*; *Kolmogorov et al., 1991*) and since then treated extensively, e.g. (*Murray, 2002*; *van Saarloos, 2003*). They admit a traveling wave solution – here, this corresponds to phages sweeping over an uninfected bacterial lawn. In general, the asymptotic expansion speed for the traveling wave solutions is given by the following expression,

$$\begin{aligned} v &= 2\sqrt{D(\partial_P G)[0,S]}\\ &= 2\sqrt{D\delta^\star}\sqrt{S\beta - 1 - \lambda\alpha}\,. \end{aligned}\tag{27}$$

Only the linearized growth rate of phages at very low densities is relevant for the expansion speed, $\partial_P G[P=0,S]$. Thus, the fraction of susceptible individuals $S$ should be unchanged from its initial value $S_0$. It should be noted, that only for $S_0\beta > 1 + \lambda\alpha$ does *Equation 27* remain valid, otherwise we have $v = 0$. Such a scenario is relevant when nutrients are depleted and phage growth parameters changes to $\beta_{\mathrm{depl}}$ and $\lambda_{\mathrm{depl}}$.

The expression for the expansion speed also shows the need for the spatial adsorption model in *Equation 24*, in contrast to the liquid case *Equation 13*. If adsorption would directly depend on the bacterial density $B$, the additional linear dependence on $B$ in *Equation 26* would lead to an exponentially increasing speed during the experiment. This is in clear contradiction to experimental observations.

The density of phages behind the expanding front is large and as previously noted at large MOIs the CRISPR-Cas system fails to provide effective immunity (see section Materials and methods and appendix Infection load and efficiency of the CRISPR/Cas system). However, in comparison to an unstructured environment (e.g., liquid) the structured environment effectively limits transit of phage from within a plaque to the expanding front: The combined effect of growth and diffusion usually generates a much faster expansion of phages during plaque formation, than diffusion alone. Only when nutrients are depleted, can pure diffusion processes explain the slow decrease in speed observed in experiments (see *Figure 7A*). Our model assumes a sharp drop to $v = 0$ at $T_{\mathrm{depl}}$ for small $S$.

In order to derive an expression for the plaque radius $r$, we integrate the expansion speed *Equation 7* over time, $r(t) = \int_0^t dt'v(t')$. Employing the simplification that only two values of phage growth are necessary to describe the dynamics – before $T_{\mathrm{depl}}$ phages grow normally with $\beta$ and $\lambda$, after $T_{\mathrm{depl}}$ phage growth changes to $\beta_{\mathrm{depl}}$ and $\lambda_{\mathrm{depl}}$ – we can evaluate the integral for the radius directly, arriving at,

$$r(t) = \begin{cases} 2t\sqrt{D\delta^\star}\sqrt{S\beta - 1 - \lambda\alpha}\,, & 0 < t < T_{\mathrm{depl}}\,,\\ 2\sqrt{D\delta^\star}\Big(T_{\mathrm{depl}}\sqrt{S\beta - 1 - \lambda\alpha} + \big(t - T_{\mathrm{depl}}\big)\sqrt{S\beta_{\mathrm{depl}} - 1}\Big)\,, & T_{\mathrm{depl}} < t\,. \end{cases}\tag{28}$$

Using this expression we estimated the adsorption constant $\delta^\star$ from the growth experiments as it difficult to measure in practice. This estimate is done for radii exactly at the time of nutrient depletion $T_{\mathrm{depl}}$, and excluding the control experiment with only susceptible cells.

Predictions of our model show a discrepancy from experimental results on plates after depletion. We independently estimated $\beta_{depl} = 3.0$, which results in $H_{depl} = (\beta_{depl} - 1)/\beta_{depl} \approx 0.67$. Thus, all experiments with $S > 0.33$ should exhibit expanding plaques after nutrients are depleted. In the experimental setup plaques stop expanding in all mixtures of resistant to susceptible cells ($S \leq 0.9$), which would correspond to $\beta_{depl} < 1.1$. This value is, however, still within experimental accuracy of our estimates of $\beta_{depl}$.

## Acknowledgements

We are grateful to Remy Chait for his help and assistance with establishing our experimental setups and to Tobias Bergmiller for valuable insights into some specific experimental details. We thank Luciano Marraffini for donating us the pCas9 plasmid used in this study. We also want to express our gratitude to Seth Barribeau, Andrea Betancourt, Călin Guet, Mato Lagator, Tiago Paixão and Maroš Pleška for valuable discussions on the manuscript. Finally, we would like to thank the eLife editors and reviewers for their helpful comments and suggestions.

## Additional information

### Funding

| Funder | Grant reference number | Author |
| --- | --- | --- |
| H2020 European Research Council | EVOLHGT No. 648440 | Jonathan P Bollback |

The funders had no role in study design, data collection and interpretation, or the decision to submit the work for publication.

### Author contributions

Pavel Payne, Conceptualization, Data curation, Software, Formal analysis, Validation, Investigation, Visualization, Methodology, Writing—original draft, Project administration, Writing—review and editing; Lukas Geyrhofer, Conceptualization, Data curation, Software, Formal analysis, Visualization, Methodology, Writing—original draft, Writing—review and editing; Nicholas H Barton, Conceptualization, Resources, Supervision, Methodology, Project administration, Writing—review and editing; Jonathan P Bollback, Conceptualization, Resources, Supervision, Funding acquisition, Validation, Methodology, Project administration, Writing—review and editing

### Author ORCIDs

Pavel Payne http://orcid.org/0000-0002-2711-9453
Lukas Geyrhofer http://orcid.org/0000-0002-8043-2975
Jonathan P Bollback https://orcid.org/0000-0002-4624-4612

### Decision letter and Author response

Decision letter https://doi.org/10.7554/eLife.32035.034
Author response https://doi.org/10.7554/eLife.32035.035

## Additional files

### Supplementary files

• Transparent reporting form
DOI: https://doi.org/10.7554/eLife.32035.021

### Major datasets

The following dataset was generated:

| Author(s) | Year | Dataset title | Dataset URL | Database, license, and accessibility information |
|-----------|------|---------------|-------------|------------------------------------------------|
| Payne P, Geyrhofer L, Barton NH, Bollback JP | 2017 | Data from: CRISPR-based herd immunity limits phage epidemics in bacterial populations | https://doi.org/10.5061/dryad.42n44 | Available at Dryad Digital Repository under a CC0 Public Domain Dedication |

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

## Appendix 1

DOI: https://doi.org/10.7554/eLife.32035.022

# Additional theoretical considerations

## Simulation of recovery rate

Throughout the main text we assumed that resistant bacteria are completely immune to phage infection as their CRISPR/Cas system immediately kills adsorbed phages. However, experimental observation suggest that for fractions close to what we predicted as herd immunity threshold, all bacteria eventually die. Thus, in the following section we use numerical simulations to investigate the full set of equations *Equation 12*, with a particular focus on the question why the whole bacterial population goes extinct. As it turns out, this requires using finite values for the recovery rate $\rho$ (instead of the $\rho \to \infty$ approximation employed previously).

A major difficulty in analyzing the full model *Equation 12* is finding appropriate parameter values. In particular, we need values for the adsorption constant $\delta$, the recovery rate $\rho$ and the yield coefficient $Y$. Undiluted LB medium is known to support a population of $5 \cdot 10^9$ cells/ml. Thus one can easily estimate $Y$ as the inverse of this number, when nutrients are measured in units of dilutions, which we already used throughout this publication (undiluted medium corresponds to $N = 1$). Parameter scans in simulations reveal that the actual value of the adsorption constant $\delta$ does usually not influence the actual outcome (collapsed or surviving bacterial population), it only adjusts time scales. However, deviations in time scales are insignificant, even when $\delta$ is changed by orders of magnitude, $\delta \sim \mathcal{O}(10^{-6} \ldots 10^{-8})$. They are roughly an hour or less, which is small compared to the expected duration of the experiment that lasts a few hours. For definiteness, we use the value of $\delta = 10^{-7} \ h^{-1}$ for our simulations. That the value of the adsorption constant has only a minor impact on phage growth on bacterial cultures, is also in line with previous findings (*Mitarai et al., 2016*).

The most elusive parameter is the recovery rate $\rho$. A first indication of the value of $\rho$ can be drawn from our experiments on bursting resistant cells, summarized in *Figure 2*. As the probability for bursting resistant cells is 3 orders of magnitude smaller than for susceptible bacteria, we can use $1/\lambda \sim \mathcal{O}(1)$ to estimate $\rho \sim \mathcal{O}(10^3)$. However, our results also indicate that recovery via the CRISPR/Cas system heavily depends on MOI, implying that $\rho$ depends on the actual densities of phages and bacteria. Nevertheless, as experimental determination of recovery is complicated, even more so determining a functional dependence on dynamically changing densities $B$ and $P$, we assume that $\rho$ is constant.

We ran parameter sweeps in simulations and compared the outcome – collapsed or surviving bacterial populations – to the observed experimental results (see *Figure 3*). The best agreement of simulations and experiments was reached with $\rho \sim \mathcal{O}(1)$. Lower values of $\rho$ do not allow the resistant population to recover from phage infection, while for larger values of $\rho$, phages are drained from the culture very fast. Such a small value of $\rho$ is most likely related to the recovery at very large MOI, when the densities involved in the dynamics are large, which dominate the overall observed dynamics. At this time phages repeatedly infect the same bacteria and their CRISPR/Cas immune system cannot deal with such an infection load (or only too slow). Thus, we can argue that our final choice $\rho = 1.5 \ h^{-1}$ is the recovery rate when the CRISPR/Cas system is heavily stressed, which is comparable to the actual burst rate $1/\lambda$ for phages.

In *Appendix 1—figure 1* we show three exemplary sets of trajectories for bacteria and phage. For a tiny fraction of susceptibles, $S = 10^{-3}$, which is well below the herd immunity threshold (see *Figure 3*), phages do not thrive on the limited number of favorable hosts and decay fast after a slight increase initially. For intermediate fractions of susceptibles, $S = 0.04$, we observe more complex, non-monotonic trajectories of bacterial populations. For such values of $S$ we also observe mixed outcomes in experiments, see *Figure 3*. When $S$ is increased further ($S = 0.06$), enough susceptible bacteria exists to produce enough phages and ultimately the whole bacterial population goes extinct.

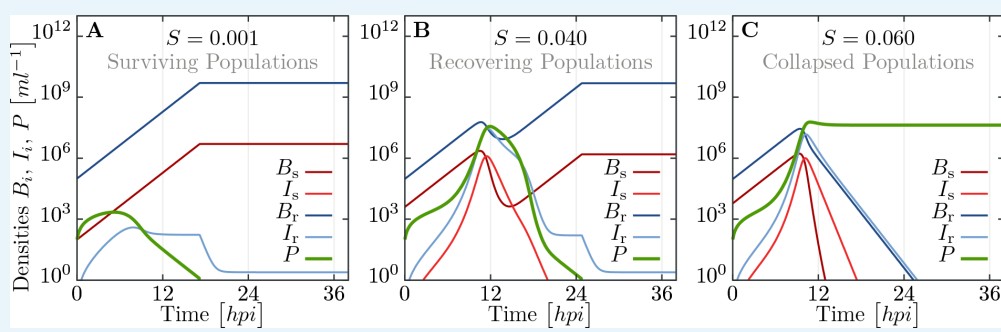

**Appendix 1—figure 1.** Simulated trajectories for all populations in liquid culture for the extended model, including infected and recovering bacteria. Trajectories are obtained by numerically integrating equations *Equation 12*, using parameters listed in *Appendix 1—table 1* and additionally $N = 1$, $Y = 2 \cdot 10^{-10}$ cells$^{-1}$, $\delta = 10^{-7}$ $h^{-1}$ and $\rho = 1.5$ $h^{-1}$. (A) For population compositions with a large majority of resistant cells ($S = 10^{-3}$), phages get wiped out fast. (B) For intermediate $S$ (close to parameters where we observe both, collapsed and surviving, populations, see *Figure 3*), the populations exhibit a complex, non-monotonic trajectory. After fast initial growth of phages, bacterial populations decay but ultimately can recover. (C) If the fraction of susceptibles is too large ($S = 0.06$), the whole bacterial population is infected and succumbs to the overwhelming phage infection. See supporting text for more detailed information.

DOI: https://doi.org/10.7554/eLife.32035.023

**Appendix 1—table 1.** Parameters used in simulations shown in *Appendix 1—figure 1*.

| Parameter | | Value | Units | Comment |
|---|---|---|---|---|
| Bacterial growth rate | $a$ | 0.63 | $1/h$ | *Table 1*, *Figure 4* |
| Yield | $Y$ | $2 \cdot 10^{-1}$ | $1/$cell | Measured in dilution of LB |
| Recovery rate | $\rho$ | 1.5 | $1/h$ | See this appendix |
| Adsorption constant | $\delta$ | $10^{-7}$ | $1/(h\,\text{cell})$ | See this appendix |
| Diffusion constant | $D$ | $1.17 \cdot 10^{-2}$ | $mm^2/h$ | See Methods |
| Burst size | $\beta$ | 85.6 | phages/cell | *Table 2*, *Figure 5* |
| Latency time | $\lambda$ | 0.60 | $h$ | *Table 2*, *Figure 5* |
| Initial bacterial population | $B_0$ | $10^5$ | cells | |
| Initial phage population | $P_0$ | 10 | phages | |

DOI: https://doi.org/10.7554/eLife.32035.024

The purpose of the extended model in this section was to justify the fact that phages can wipe out the whole bacterial population, which was not possible in the simplified model used in the main text. There, the resistant bacterial population was basically unaffected by phages and just acted as 'sink' for phages. However, also in this extended model, we see a very similar behavior in terms of the threshold phenomena reported earlier in the manuscript.

## Infection load and efficiency of the CRISPR/Cas system

In the section Modelling we showed that positive phage growth leads eventually to a very fast increase in the phage population, that occurs before nutrients are depleted (for almost all realistic parameters). This behavior of the dynamics was also observed in the extended simulation model presented in the last section. Moreover, as a condition we used that the phage population reaches a size $P \sim 1/\delta$, which is after all arbitrary – it only determines if we can employ useful simplifications and approximations to model equations. However,

simulation results presented in the last section indicate that the bacterial population starts to decay soon after such a threshold $P \sim 1/\delta$ is exceeded.

In order to proceed, we investigate the system at time $T_\delta$ further. We assume that the phage population is large enough that it will not be degraded by the CRISPR/Cas immune system. The threat to immediate phage extinction is low at this point. The actual equations are hard to solve directly, hence we revert to simple balance equations, ignoring the dynamical component. Specifically, we compare the number of (present and eventually produced) phages to the number of infections needed to wipe out the whole population. To incorporate the effects of the bacterial immune system in resistant bacteria, we assume that they need $M > 1$ infections before they burst and produce only $\kappa\beta$ phages, which reduces the burst size by a (yet unspecified) factor $0 < \kappa < 1$. $\kappa = 1$ implies that resistant cells produce the same number of phages as susceptible cells, while $\kappa = 0$ indicates only cell death. Combining these considerations yields

$$\underbrace{1/\delta}_{\text{phages} \sim \text{present}} + \underbrace{\beta S_0 B(T_\delta)}_{\text{phage} \sim \text{production} \sim B_s} + \underbrace{\kappa\beta(1-S_0)B(T_\delta)}_{\text{phage} \sim \text{production} \sim B_r} > \underbrace{S_0 B(T_\delta)}_{\text{infections} \sim B_s} + \underbrace{M(1-S_0)B(T_\delta)}_{\text{infections} \sim B_r}, \qquad (29)$$

where the left side indicates the total number of phages, while the right side indicates the number of necessary infections to kill all bacteria. The number of bacteria $B(T_\delta)$ can be estimated by inserting the time $T_\delta$ from **Equation 18** into the exponential growth **Equation 15b**. Subsequently, we can rearrange **Equation 29**, obtaining a bound on $M$:

$$M < \frac{1/\delta B(T_\delta) + S_0(\beta - 1)}{1 - S_0} + \kappa\beta . \qquad (30)$$

The first term $1/\delta B(T_\delta)$ indicates the ratio of phages to bacteria at time $T_\delta$, and can be considered small for non-extremal parameters compared to the other terms. This fact justifies our assumption that the actual value of $\delta$ is not crucial. This number $M$ might allow some insight into the effectiveness of the CRISPR/Cas immune system. For a fraction of susceptibles $S = 0.03$, which corresponds to the minimal value where we observe only collapsed bacterial populations in undiluted LB medium (see **Figures 3,5**), we would obtain the relation $M \lesssim 3 + 86\kappa$. Thus, each resistant bacterial cell could degrade up to $\mathcal{O}(10^1 \ldots 10^2)$ phages before their CRISPR/Cas system cannot cope with the infection load anymore.

# Appendix 2

DOI: https://doi.org/10.7554/eLife.32035.025

## Supplementary results

### Reduction in number of plaques in spatially structured populations

The reduction in the number of plaques with increasing proportions of resistant bacteria is shown in *Appendix 2—figure 1*.

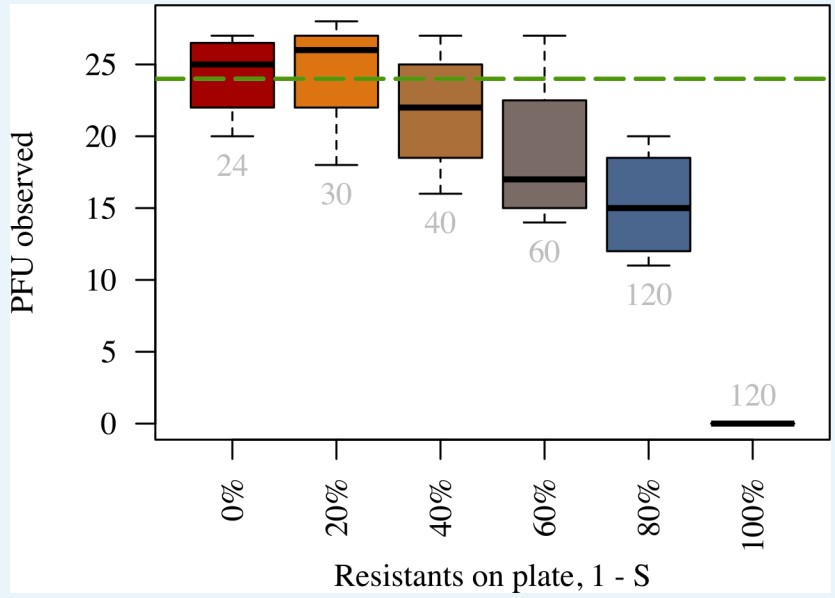

**Appendix 2—figure 1.** Number of plaques declines faster than proportionally to the fraction of resistant bacteria. Number of plaque forming units observed on a plate (y-axis) for different proportions of resistant bacteria (x-axis). Grey numbers below each boxplot indicate the average number of phages inoculated in the respective treatment. The numbers of phages inoculated were chosen to retain the expected number of plaques on the plate (green dashed line) as in the 0% resistant treatment (red boxplot). Plates were prepared using identical procedure as in Time-lapse imaging of plaque growth (see Materials and methods). The data presented in this figure can be found in *Appendix 2—figure 1—source data 1*.

DOI: https://doi.org/10.7554/eLife.32035.026

The following source data is available for figure :
**Appendix 2—figure 1—source data 1.** Measurements of plaque numbers in populations consisting of varying proportions of resistant to susceptible bacteria.

DOI: https://doi.org/10.7554/eLife.32035.027

## Appendix 3

DOI: https://doi.org/10.7554/eLife.32035.028

### Additional information on experimental setup

Our experimental setup for the scanner system is shown in *Appendix 3—figure 1*.

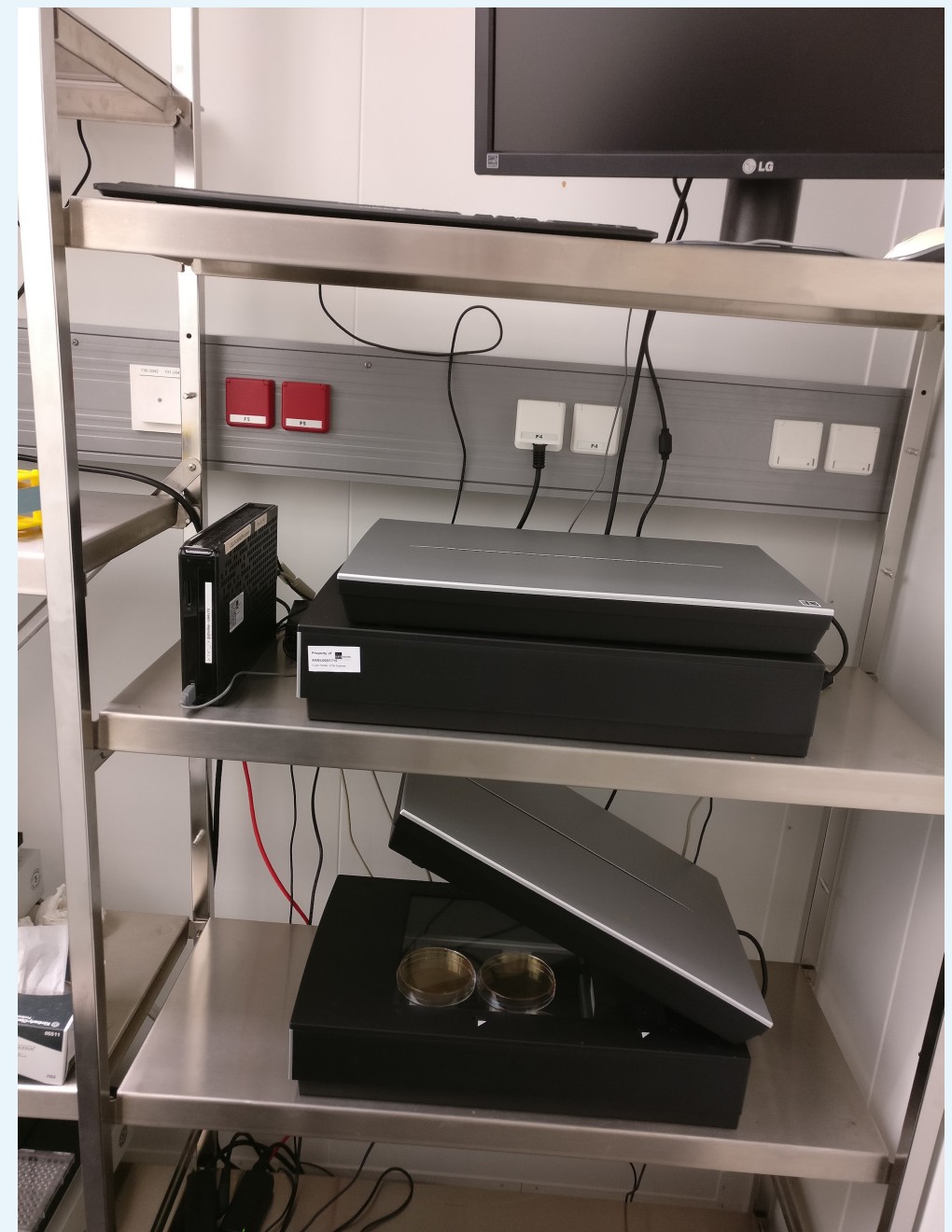

**Appendix 3—figure 1.** Image of the scanner system. Photograph of the scanner system used for time-lapse imaging of phage spread in spatially structured bacterial populations. Three scanners (Epson Perfection V600 Photo Scanner) simultaneously scanned 12 plates in total every 20 min in $30°C$ for 48 hr per experiment.

DOI: https://doi.org/10.7554/eLife.32035.029

## Appendix 4

DOI: https://doi.org/10.7554/eLife.32035.030

### Source data and code

Time-lapse images of spread of T7 phage epidemics in Escherichia coli spatially structured populations are available on the Dryad Digital Repository: https://doi.org/10.5061/dryad.42n44. Source code of the model presented here is available on GitHub (https://github.com/lukasgeyrhofer/phagegrowth) (*Payne et al., 2018*) and its archived version is accessible through Zenodo: https://dx.doi.org/10.5281/zenodo.1038582 and https://github.com/elifesciences-publications/phagegrowth.

