## [Decision Letter]

Thank you for submitting your article "CRISPR-based herd immunity limits phage epidemics in bacterial populations" for consideration by *eLife*. Your article has been reviewed by three peer reviewers, one of whom Richard A Neher is a member of our Board of Reviewing Editors and the evaluation has been overseen by and Arup Chakraborty as the Senior Editor. The following individuals involved in review of your submission have agreed to reveal their identity: Timothy Cooper (Reviewer #2).

The reviewers have discussed the reviews with one another and the Reviewing Editor has drafted this decision to help you prepare a revised submission.

Summary:

This manuscript presents a mathematical model and a set of experiments examining how CRISPR-Cas mediated immunity to phage infection can result in herd immunity preventing the spread of the pathogen. The experiments are performed both in well-mixed liquid culture as well as in structured environments on agar. The critical herd immunity thresholds and the plaque radii follow model predictions as the parameters of the system as varied. All reviewers appreciated the elegant combination of experiments and theory, but we would like the authors to address the following points:

Essential revisions:

1) The system is set up such that resistant cells don't grow once infected. We would like to see the importance of this assumption discussed. How would model predictions change if cells kept growing? How important is rapid degradation of phage by resistant cells?

2) We would like to encourage the authors to move some derivations presented in Materials and methods section to the Results section. While detailed algebra is better kept in the Materials and methods section, additional mathematical details would help readers appreciate the theoretical results without having to jump back and forth to the Materials and methods section.

3) The experiments are started with a small inoccula which result in a broad crossover at the herd immunity threshold. Is the observed stochasticity consistent with that expected from a small founder population? Or are there additional sources of stochasticity? Would the transition in Figure 4C be sharper if larger inoccula had been used? What is the nature of the error bars in Figure 3 and Figure 4C? Do they quantify uncertainty in the parameter measurements or the stochasticity of the dynamics?

4) There is a sophisticated body of theory on pathogen spreading with long range jumps (e.g. Hallatschek and Fisher, 2014). Can you provide some intuition/discussion how the spreading and herd immunity thresholds change as dispersal goes from 2D diffusion to occasional long-range jumps ultimately to the well-mixed case?

5) It is mentioned in the Abstract that herd immunity might facilitate coexistence between susceptible and resistant variants. This possibility is mentioned again in the discussion in one sentence but not elaborated on. While it seems obvious that selection for resistant variants is reduced when the pathogen can no longer spread due to herd immunity, it is not obvious that this herd immunity results in stable coexistence. Resistant variants would still sweep to high frequencies in areas with high pathogen load. How would a cost of resistance change the model behavior?

The quantitative model the authors parameterized with experiments could be used to investigate many of these points in greater detail. If the model for example predicted the width of the cross-over at the herd immunity threshold correctly, such comparisons would increase confidence in both the model and the experimental system. Furthermore, the model could be used to assess robustness to assumptions like growth arrest and rapid phage degradation.

---

## [Author Response]

Essential revisions:1) The system is set up such that resistant cells don't grow once infected. We would like to see the importance of this assumption discussed. How would model predictions change if cells kept growing? How important is rapid degradation of phage by resistant cells?

We agree with the reviewers that this should have been discussed in more detail. Therefore, we added to the Results section a discussion of the importance of this property of our experimental system – namely, that resistant bacteria stop growing once infected.

Degradation of the phage by resistant bacteria is, in our opinion, an important pre-requisite of (bacterial) herd immunity. If the phage particles survive an encounter with a resistant bacterium (e.g. with a mutated receptor protein to which phages cannot adsorb) and can go on to infect other individuals, they would have to decay in the environment at the timescale comparable to their adsorption rate in order for resistant bacteria to provide any indirect protection to the susceptible ones. We argue that the usual persistence of the phage particles in both laboratory and natural environments must be substantially longer as dwelling outside the host likely covers a majority of phage life history. Also, common laboratory experience shows that titres of many different phage species remain stable over very large time periods in a variety of media and buffers. In order to clarify this point, we comment on the potential degradation of phage particles in the environment in the Introduction.

2) We would like to encourage the authors to move some derivations presented in Materials and methods section to the Results section. While detailed algebra is better kept in the Materials and methods section, additional mathematical details would help readers appreciate the theoretical results without having to jump back and forth to the Materials and methods section.

We have brought Equations 24 and 26 in a combined form, as well as their description, into the Results section to aid readers' understanding of the model. Both equations remain in the Materials and methods section to aid with understanding the derivations. In addition, we have added an explanation in the Results section of how the plaque radius was derived using the Equation 7 for the speed of expansion and Equation 3 for nutrient levels (subsection “Modelling herd immunity in spatially structured populations”). We also reformulated some text in the modelling section of the Results section and of the Materials and methods section for a better understandability.

3) The experiments are started with a small inoccula which result in a broad crossover at the herd immunity threshold. Is the observed stochasticity consistent with that expected from a small founder population? Or are there additional sources of stochasticity? Would the transition in Figure 4C be sharper if larger inoccula had been used? What is the nature of the error bars in Figure 3 and Figure 4C? Do they quantify uncertainty in the parameter measurements or the stochasticity of the dynamics?

We have now added a paragraph in the Results section explaining in detail the sources of stochasticity, the nature of the error bars in Figure 3 and Figure 4C and have commented on the potential effect of a larger phage inocula.

4) There is a sophisticated body of theory on pathogen spreading with long range jumps (e.g. Hallatschek and Fisher, 2014). Can you provide some intuition/discussion how the spreading and herd immunity thresholds change as dispersal goes from 2D diffusion to occasional long-range jumps ultimately to the well-mixed case?

In general, we would not expect that long range jumps change the herd immunity threshold H(α) itself. Spread of pathogens still stops when the fraction of susceptible hosts S is small such that βS < 1 + λα, and will continue as long as βS > 1 + λα is fulfilled. However, we expect that long-range jumps of phages will enhance the overall speed of expansion, and significantly increase the area covered by the phage infection. In order to address these concerns, we have added a paragraph (subsection “Modelling herd immunity in spatially structured populations”) providing some intuition of how long-range jumps might affect epidemic spread and the herd immunity threshold, in both general terms and in our experimental system.

5) It is mentioned in the Abstract that herd immunity might facilitate coexistence between susceptible and resistant variants. This possibility is mentioned again in the discussion in one sentence but not elaborated on. While it seems obvious that selection for resistant variants is reduced when the pathogen can no longer spread due to herd immunity, it is not obvious that this herd immunity results in stable coexistence. Resistant variants would still sweep to high frequencies in areas with high pathogen load. How would a cost of resistance change the model behavior?

We have elaborated on the role of herd immunity in coexistence of resistant and susceptible individuals as well as the pathogen in a population in greater detail in the Discussion section. We also discussed what role a cost of immunity may play in the dynamics of herd immunity and coexistence of the pathogen and the hosts.

The quantitative model the authors parameterized with experiments could be used to investigate many of these points in greater detail. If the model for example predicted the width of the cross-over at the herd immunity threshold correctly, such comparisons would increase confidence in both the model and the experimental system. Furthermore, the model could be used to assess robustness to assumptions like growth arrest and rapid phage degradation.